# Hidden diversity and potential ecological function of phosphorus acquisition genes in widespread terrestrial bacteriophages

Jie-Liang Liang [1], Shi-wei Feng[1], Jing-li Lu[1], Xiao-nan Wang[1], Feng-lin Li[1], Yu-qian Guo[1], Shen-yan Liu[1], Yuan-yue Zhuang[1], Sheng-ji Zhong[1], Jin Zheng[1], Ping Wen[1], Xinzhu Yi[1], Pu Jia[1], Bin Liao[2], Wen-sheng Shu [1] & Jin-tian Li [1] ✉

Phosphorus (P) limitation of ecosystem processes is widespread in terrestrial habitats. While a few auxiliary metabolic genes (AMGs) in bacteriophages from aquatic habitats are reported to have the potential to enhance P-acquisition ability of their hosts, little is known about the diversity and potential ecological function of P-acquisition genes encoded by terrestrial bacteriophages. Here, we analyze 333 soil metagenomes from five terrestrial habitat types across China and identify 75 viral operational taxonomic units (vOTUs) that encode 105 P-acquisition AMGs. These AMGs span 17 distinct functional genes involved in four primary processes of microbial P-acquisition. Among them, over 60% (11/17) have not been reported previously. We experimentally verify in-vitro enzymatic activities of two pyrophosphatases and one alkaline phosphatase encoded by P-acquisition vOTUs. Thirty-six percent of the 75 P-acquisition vOTUs are detectable in a published global topsoil metagenome dataset. Further analyses reveal that, under certain circumstances, the identified P-acquisition AMGs have a greater influence on soil P availability and are more dominant in soil metatranscriptomes than their corresponding bacterial genes. Overall, our results reinforce the necessity of incorporating viral contributions into biogeochemical P cycling.

The widespread occurrence of phosphorus (P) limitation of ecosystem processes has been documented not only in aquatic habitats but also in terrestrial habitats[1,2]. Only a small proportion (<6%) of total soil P content is bioavailable[3], and therefore many biological processes in terrestrial ecosystems are constrained by low soil P availability[4]. Especially, there is increasing evidence that low soil P availability strongly limits microbial processes in a variety of terrestrial ecosystems around the world[4,5].

Enhancing P-acquisition ability is a key strategy used by microorganisms to cope with P scarcity in their habitats[6]. Some microbes harbor diverse metabolic capacities to improve the bioaccessibility of various recalcitrant P forms in environments[6–8]. The ability of such microbes to acquire P from the environment is critical not only for themselves but also for other components in the ecosystem. For instance, phosphate-solubilizing microbes can mobilize P from sparingly available P forms and thereby facilitate the plant P uptake[7,9]. Therefore, microbes have an important influence on P dynamics in environments, and exploring the mechanisms of microbial P-acquisition (especially under P-limited conditions) is essential to a better understanding of biogeochemical P cycling.

On the one hand, microorganisms can enhance P bioavailability through direct mineralization and solubilization of recalcitrant P by

[1]Institute of Ecological Science, Guangzhou Key Laboratory of Subtropical Biodiversity and Biomonitoring, Guangdong Provincial Key Laboratory of Biotechnology for Plant Development, School of Life Sciences, South China Normal University, Guangzhou 510631, PR China. [2]School of Life Sciences, Sun Yatsen University, Guangzhou 510275, PR China. ✉e-mail: lijintian@m.scnu.edu.cn

the release of hydrolytic enzymes[10]. There are a few microbial enzymes able to release free orthophosphate from recalcitrant organic P forms, such as acid phosphatase (encoded by *aphA*, *olpA*, or *phoN*), alkaline phosphatase (*phoA*, *phoD*, or *phoX*), C-P lyase (*phnP*), glycerophosphoryl diester phosphodiesterase (*ugpQ*), and phosphoribosyl 1,2-cyclic phosphate phosphodiesterase (*phnW*)[10,11]. Also, there are two proteins involved in the hydrolysis of inorganic phosphate polymers, namely, inorganic pyrophosphatase (PPa, encoded by *ppa*) and exopolyphosphatase (*ppx*)[8].

On the other hand, microorganisms have to compete with other biota for limited available P in environments. Therefore, they often harbor efficient P uptake/transport and P starvation response systems. Orthophosphate in environments can be transported into microbial cells by permease proteins encoded by a high-affinity phosphate-specific transport system (Pst, encoded by *pstS*, *pstC*, *pstA*, and *pstB*) and/or a low-affinity inorganic phosphate transport system (PiT, *pit*), while glycerol-3-phosphate is transported by proteins encoded by the Ugp system (*ugpB*, *ugpA*, *upgE*, and *upgC*)[8]. The phosphate regulon (Pho) is a regulatory mechanism involved in the sensing and regulation of available phosphate, controlled by a two-component regulatory system known as PhoR-PhoB (*phoR* and *phoB*)[12]. The Pst system is a highly conserved component of the Pho regulon[13]. Besides the Pst system, *phoH*, a gene that encodes a putative ATPase, is found in the Pho regulon of bacteria, although its precise role remains unclear[14]. PhoU (*phoU*) can act as a negative regulator of the Pho regulon[15].

Intriguingly, several lines of evidence implicate that, in P-limited aquatic ecosystems, certain phages (i.e., viruses that infect bacteria) have the potential to facilitate P uptake of their hosts by encoding auxiliary metabolic genes (AMGs) involved in microbial P-acquisition. Nine cyanophages isolated from low-P oceans were found to contain a periplasmic high-affinity phosphate-binding gene *pstS*, and two of these cyanophages also encoded an alkaline phosphatase gene *phoA*[14]. Further metagenomics and metatranscriptomics revealed that P stress was a strong selective pressure for oceanic phages to retain *pstS*[16,17]. Recently, several AMGs involved in microbial P-acquisition (e.g., *ppa*, *phoD*, and *yjbB*) have proven to be expressed in surface waters and sediments of the Pearl River estuary with a low level of available P[18]. In addition, some viral sequences from P-deficient acid mine drainage sediments were reported to harbor *phoH* genes[19]. Note, however, that among the currently known P-acquisition AMGs encoded by aquatic phages, *pstS* related to P transportation is the only one whose function has been validated experimentally[20].

Since P is one of the growth limiting factors for bacteria and a significant amount of P is also required for phage replication, phages possessing P-acquisition AMGs possibly gain a fitness advantage during their infection in P-limited environments[16]. Given that terrestrial and aquatic microbes tend to employ similar strategies to cope with P limitation[6], one may expect that phages carrying a variety of P-acquisition AMGs likely exist in P-limited terrestrial ecosystems. However, such phages remain poorly described. In 2022, three viral *phoD* genes were detected by Zheng et al. in industrial soils and eight viral *phoH* genes were recovered by Han et al. from agricultural soils[21,22]. More recently, Huang et al. identified six P-acquisition AMGs spanning three distinct functional genes (i.e., *phoA*, *phoB*, and *phoH*) in paddy soils[23].

In this study, we analyzed 333 soil metagenomes from five distinctive terrestrial habitat types (i.e., farmland, forest, grassland, Gobi desert, and mine wasteland; the last two of which are extremely deficient in P) across China (Fig. 1) to identify those phages with P-acquisition AMGs. We referred to such phages as 'P-acquisition

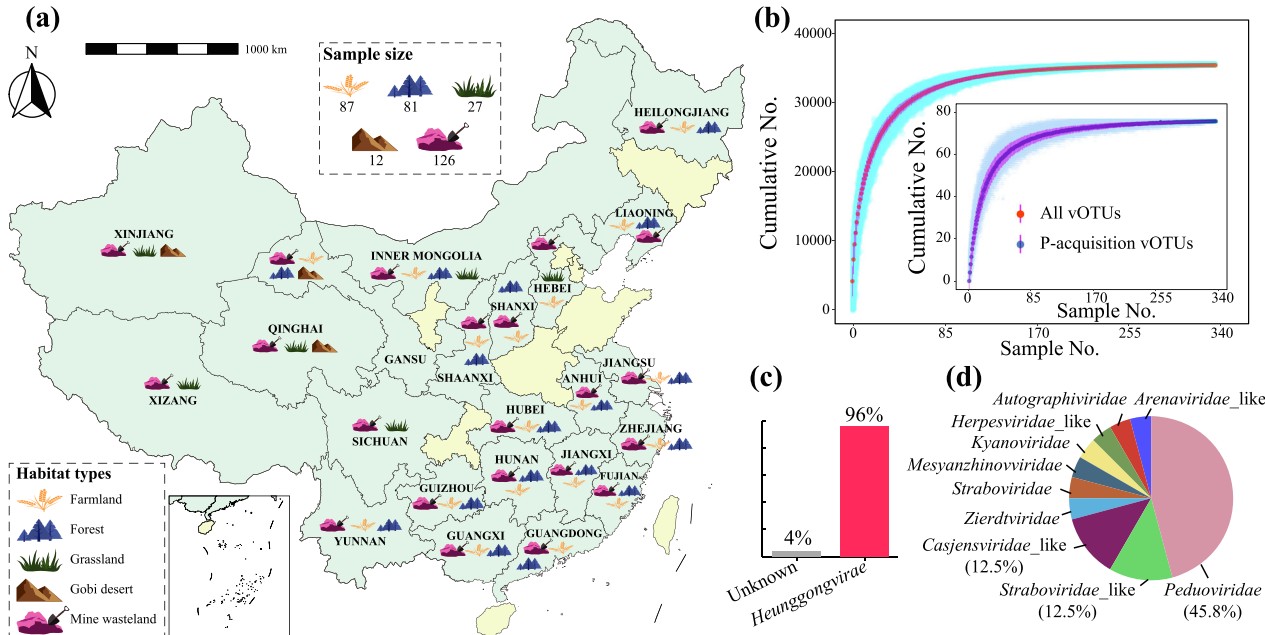

**Fig. 1 | Sampling sites and selected main characteristics of soil phages under investigation. a** Geographic illustration of the soil sampling sites of this study on the map of China. Detailed information of each sampling site is provided in Supplementary Data 1. The darker green color on the map is indicative of areas where samples were taken, and the lighter green color represents areas where no samples were collected. The map of China was obtained from http://geo.datav.aliyun.com/areas_v2/bound/100000_full.json and was visualized by the R package sf. **b** Sample-based accumulative curves of all viral operational taxonomic units (vOTUs) recovered from soil metagenomes of this study and those vOTUs encoding genes responsible for microbial P-acquisition (referred to as 'P-acquisition vOTUs', shown in the inset). The accumulative curves were generated by a custom R script (available on GitHub)[74] utilizing the R packages foreach and doParallel. Data ($n = 500$) were presented as mean values ± standard deviations. Teal or light blue traces represent 500 iterations (sample order randomizations), and red or blue points are means. **c, d** The kingdom- and family-level classifications of the P-acquisition vOTUs according to geNomad (bar chart) and PhaGCN (pie chart), respectively. For better visualization, only the percentages of the three most dominant families to the total numbers of vOTUs that could be classified at the level of family are shown. Detailed information of the vOTU taxonomy is provided in Supplementary Data 8.

phages'. The taxonomic diversity of identified P-acquisition phages and the functional diversity of their associated P-acquisition AMGs were assessed, and so were the geographic distributions of identified P-acquisition phages and AMGs at both country and global scales. The potential contribution of identified P-acquisition phages to soil P bioavailability was further evaluated using both metagenomic and metatranscriptomic data. Additional experiments were performed to validate the activities of enzymes encoded by representatives of identified P-acquisition AMGs. Our results revealed the hidden diversity and potential ecological function of P-acquisition AMGs in widespread terrestrial bacteriophages.

## Results

### P-acquisition phages and AMGs identified in soil metagenomes from across China

A total of 35,552 viral operational taxonomic units (vOTUs) were recovered from our 333 soil metagenomes (Fig. 1 and Supplementary Data 1). Using HMM search and Diamond BLASTP against the KEGG database, we identified 188 and 216 potential P-acquisition AMGs from these vOTUs, respectively. After eliminating duplicates, 257 potential P-acquisition AMGs were obtained. Among the 168 vOTUs encoding these 257 potential P-acquisition AMGs, 115 were further validated as phages by three methods (see Methods for details). To validate the potential P-acquisition AMGs of these 115 vOTUs, we used DRAM-v and identified 105 P-acquisition AMGs, which were encoded by 75 vOTUs (Supplementary Data 2–7). The cumulative curves of the number of total vOTUs and that of P-acquisition vOTUs with sample size appeared to reach a plateau, respectively (Fig. 1b). The average contig size of the

75 P-acquisition vOTUs was 39.8 kbp (with a range of 10 to 224 kbp, Supplementary Data 4).

Using GeNomad[24], up to 96% of the P-acquisition vOTUs (72/75) were classified to the *Heunggongvirae* kingdom (Fig. 1c), among which 56% (40/72) were of the *Caudoviricetes* class (Supplementary Data 8). However, only one of the *Heunggongvirae* vOTUs could be further classified into a specific viral family (i.e., *Herpesviridae*, Supplementary Data 8). Thus, the family-level and finer classifications of the *Heunggongvirae* vOTUs were performed alternatively with PhaGCN[25]. A total of 24 *Heunggongvirae* vOTUs could be further assigned to 10 families, among which *Peduoviridae* was the most dominant one (with 11 vOTUs, Fig. 1d). There were 17 *Heunggongvirae* vOTUs that could be classified at the subfamily or genus level (Supplementary Data 8).

The 105 identified P-acquisition AMGs spanned 17 distinct functional genes (i.e., 17 gene kinds), which could be assigned to four categories as per the primary processes of microbial P-acquisition (Fig. 2a): inorganic P solubilization (including *ppa* and *ppx*), organic P mineralization (*phnP*, *phnW*, *phoD*, and *ugpQ*), P transportation (*phnD*, *pit*, *pstS*, *pstA*, *pstB*, *pstC*, *ugpE*, and *yjbB*), and P regulation (*phoB*, *phoR*, and *phoU*). Within each of the four categories, the distinct AMGs with the highest number of gene sequences were *ppa* (8), *phoD* (23), *pstS* (6), and *phoR* (13), respectively. Among the 17 distinct functional genes, 11 (i.e., *ppx*, *phnP*, *phnW*, *ugpQ*, *pit*, *pstA*, *pstB*, *pstC*, *ugpE*, *phoR*, and *phoU*) have not been recognized as AMGs in previous studies. The addition of these novel kinds of AMGs to the 11 kinds reported previously (Supplementary Data 9) led to a dramatic increase (100%) of gene kinds of P-acquisition AMGs. Specifically, it expanded the gene kinds of AMGs related to inorganic P solubilization, organic P

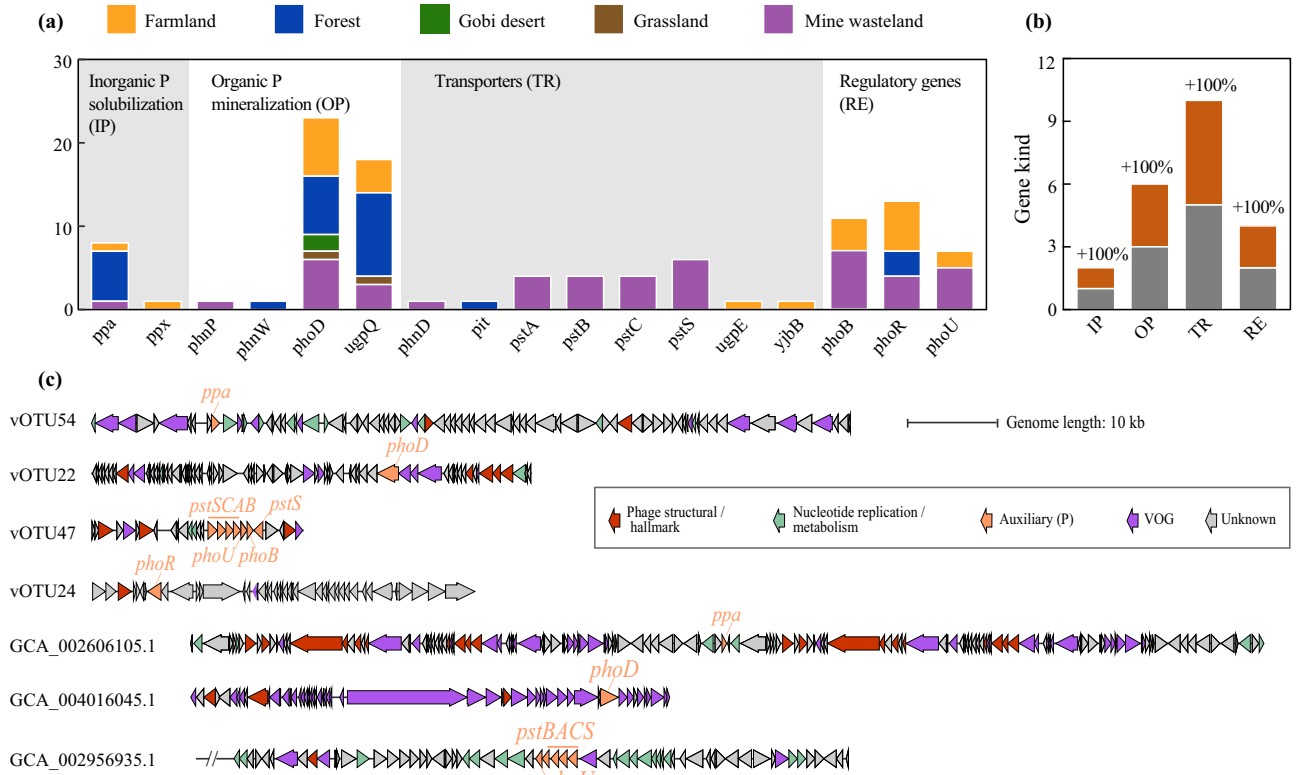

**Fig. 2 | P-acquisition auxiliary metabolism genes (AMGs) identified in this study. a** Numbers of the AMGs involved in inorganic P solubilization, organic P mineralization, P transportation, and P starvation response regulation, respectively. The numbers of each kind of AMGs recovered from individual habitat types are indicated by different colors. Detailed information on the AMGs is provided in Supplementary Data 7. **b** A graph illustrating the increase in gene kind after adding the AMGs identified in this study (orange) to those reported previously (gray; detailed information on them is provided in Supplementary Data 9). **c** Genome

organization diagrams of four representative vOTUs identified in this study (i.e., vOTU54, vOTU22, vOTU47, and vOTU24) and three representative public isolated phage genomes (i.e., GCA_002606105.1, GCA_004016045.1, and GCA_002956935.1). Due to the large genome size of GCA_002956935.1, only 30 genes upstream and downstream of the *pstBACS* cluster are displayed. Predicted open reading frames are colored according to VIBRANT and DRAM-v annotation functions. VOG, virus orthologous groups. Detail information on these viral genomes is listed in Supplementary Data 5, 6, 11.

mineralization, P transportation, and P regulation by 100%, 100%, 100%, and 100%, respectively (Fig. 2b).

## P-acquisition phages and AMGs recovered from public viral databases

In an attempt to identify other novel kinds of P-acquisition AMGs, 68,904 viral genomes were downloaded from the IGM/VR and NCBI GenBank databases to search for P-acquisition phages. A total of 106 phages were found to carry 112 P-acquisition AMGs (Supplementary Data 10). They were cultured or recovered from metagenomes from various environments, such as soil, seawater, freshwater, rhizosphere, insect, and human fecal. Among these P-acquisition phages, 98 were belonging to *Caudoviricetes*, and the others remained unclassified. As for the 112 P-acquisition AMGs, they spanned 11 gene kinds, which were overlapped by those reported previously (Supplementary Data 9) or presented in Fig. 2a. The most predominant gene kind was *pstS* (with 42 gene sequences), followed by *ugpQ* (41), *phoD* (16), *ppa* (5), *pstB* (2), *phoB* (1), *phoR* (1), *phoU* (1), *pstA* (1), *pstC* (1), and *ugpB* (1). While most of the phage *pstS* genes (40/42) were originated from marine (31) or freshwater (9) environments, nearly 70% of the phage *phoD* genes (11/16) were recovered from the soil environment (Supplementary Data 10).

## Genomic context of representative P-acquisition AMGs

Genomic arrangements of four representative P-acquisition vOTUs recovered from our metagenomes were shown in Fig. 2c. Although these vOTUs displayed diverse genomic arrangements, the associated P-acquisition AMGs were flanked by viral hallmark or viral-like genes on both sides (see Supplementary Data 5, 6 for details), supporting their affiliations to viruses. According to DRAM-v, the auxiliary scores of *ppa*, *phoD*, and *phoR* encoded by the representative P-acquisition vOTUs were 1, 2, and 2, respectively (Supplementary Data 6). Similar to our P-acquisition phages, those from public databases, such as the isolated phages GCA_002606105.1 harboring *ppa* and GCA_004016045.1 encoding *phoD* (Fig. 2c), also displayed diverse genomic arrangements (see Supplementary Data 11 for details).

While the auxiliary scores of *phoU*, *phoB*, and one *pstS* encoded by vOTU47 shown in Fig. 2c were all 2, the four genes in the *pstSCAB* cluster encoded by the same vOTU all had an auxiliary score of 4 (Supplementary Data 6). According to DRAM-v, those genes with an auxiliary score of >3 are generally not considered as AMGs. However, we found that the genome of one isolated phage from public databases (i.e., GCA_002956935.1) harbored simultaneously *phoU* and *pstBACS* in a row (Fig. 2c), providing evidence of the existence of *pst* gene cluster in a phage genome. Therefore, the members of the *pstSCAB* gene cluster encoded by vOTU47 and two additional vOTUs recovered from our metagenomes (Supplementary Data 5, 6) were considered as AMGs. Despite this, each of the 17 AMG kinds identified in our metagenomes was represented by at least one phage gene sequence with an auxiliary score of 1 or 2 (Supplementary Data 6).

## Conserved amino acid residues in proteins encoded by P-acquisition AMGs

Protein sequences of the 105 P-acquisition AMGs identified in our metagenomes were further aligned with biochemically validated reference bacterial sequences to assess the presence of conserved active residues. All protein sequences of the *ppa* and *phoR* genes had complete conserved active residues (Supplementary Figs. 1–3): PPa (including Family I and Family II), Asp residues coordinating $Mg^{2+}/Mn^{2+}$ (DxdxxD or DHH)[26]; PhoR, ATP binding (NxxxNaxky, Dxgxgi, gLa, F)[27]. The majority of protein sequences of the *phoD* (19/23) and *pstS* genes (4/6) had completely conserved active residues (Supplementary Figs. 4 and 5): PhoD, metal ligating residues coordinating $Fe^{3+}/Ca^{2+}$ (DDhe/d, DxH)[28]; PstS, phosphate binding (T, gSgxg, S/D, RxxxSgT/ D)[29]; while the rest of the PhoD and PstS protein sequences had partial

conserved active residues. Similar results were found for the other 11 kinds of AMGs (i.e., *phnD*, *phnP*, *phnW*, *phoB*, *phoU*, *pit*, *ppx*, *pstA*, *pstB*, *ugpE*, and *ugpQ*; Supplementary Figs. 6–16). Although the conserved active sites of microbial PstC and YjbB have not yet been reported, the alignment of the protein sequences of the phage *pstC* and *yjbB* genes with public references showed highly conserved regions (Supplementary Figs. 17, 18).

The structural model prediction of proteins encoded by 18 representative P-acquisition AMGs (one gene was selected for *ppa* Family I, *ppa* Family II, and each of the other 16 AMG kinds, respectively) at Phyre2 showed 100% confidence, 16%–61% identity, and 47%–98% coverage (Fig. 3, Supplementary Fig. 19, and Supplementary Data 12). The active sites were located on most of the modeled structures, except those of PstC and YjbB. Specifically, the modeled PstC structure had only 19% identity and 61% coverage with that of the template—an ABC transporter (LpqY-SugABC) that translocates trehalose (Protein Data Bank ID: 7CAF). The modeled structure of YjbB had only 20% identity and 47% coverage with that of the template—a human citrate transporter (NaCT, Protein Data Bank ID: 7JSK). The active sites of the two protein templates may be quite different from those of our proteins.

## Selective pressures on P-acquisition AMGs

To estimate the selective pressures on P-acquisition AMGs, we attempted to calculate the ratios of non-synonymous to synonymous nucleotide differences ($dN/dS$) of all 17 kinds of P-acquisition AMGs identified in our metagenomes. However, only eight kinds of these AMGs (i.e., *phoD*, *phoR*, *phoU*, *ppa*, *pstA*, *pstB*, *pstS*, and *ugpQ*) met the criteria for calculating a meaningful $dN/dS$ ratio (see Methods for details). The $dN/dS$ ratios of the eight AMG kinds were lower than 1 (Supplementary Data 13), suggesting that the phages possessing these AMGs would selectively retain the AMG's function by eliminating deleterious mutations.

## Functional validation of phage ppa and phoD

To test whether the identified P-acquisition AMGs can encode functionally active proteins, representative phage genes that were involved in inorganic P solubilization (*ppa*) or organic P mineralization (*phoD*) were selected for functional validation via in vitro assays. The functional validation of AMGs related to P transportation and P regulation (e.g., *pstS* and *phoR*, respectively) was not considered in this study for two reasons. First, the function of a phage *pstS* has been validated previously[21]. Second, the function of a regulatory gene (e.g., *phoR*) is very difficult to verify in an in vitro system, given that it involves the purification of both PhoR and PhoB proteins, and the verification of the autophosphorylation of PhoR and the phosphorylation of PhoB simultaneously.

Among the eight PPa proteins encoded by AMGs identified in our metagenomes, one (i.e., PPa1 encoded by vOTU1) belonged to the soluble PPa Family I and the others (e.g., PPa54 encoded by vOTU54) were of the soluble PPa Family II (Supplementary Fig. 20). Therefore, two phage *ppa* genes (i.e., *ppa*1 and *ppa*54) were cloned, expressed in *E. coli*, and then functionally assayed, respectively (Fig. 3 and Supplementary Fig. 21). Our results demonstrated that, both PPa proteins were capable of hydrolyzing pyrophosphate into phosphate. The enzyme activities of PPa1 and PPa54 were 500 and 1005 U mL$^{-1}$, respectively (Fig. 3b, f). Remarkably, the enzyme activity of PPa54 was comparable to that of the commercial *Escherichia coli* PPa (1000 U mL$^{-1}$). The optimal pHs of the two PPa proteins were 8.0 (PPa1) and 7.5 (PPa54), respectively (Fig. 3c, g), while their optimal temperatures were 40 °C and 37 °C correspondingly (Fig. 3d, h).

The PhoD22 encoded by vOTU22 had highly conserved active residues (Supplementary Fig. 4), although its protein length was predicted to be longer than those of some known bacterial PhoD proteins[28]. The *phoD*22 was cloned, expressed in *E. coli*, and then

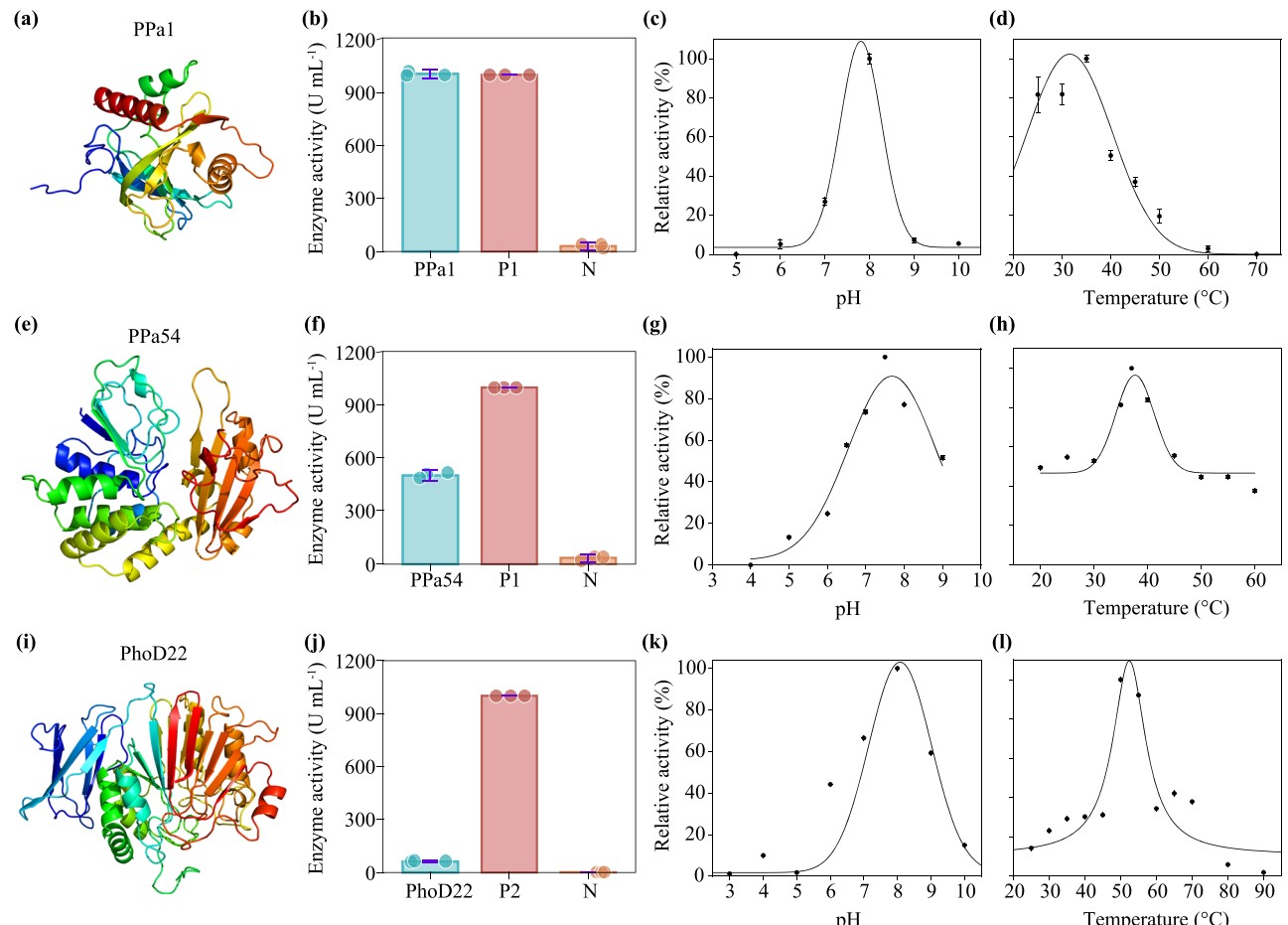

**Fig. 3 | Functional validation of three representative P-acquisition AMGs.**
**a**, **e**, **i** show the computational protein models of two pyrophosphatases (i.e., PPa1 encoded by the AMG *ppa*1 and PPa54 encoded by the AMG *ppa*54) and one alkaline phosphatase (i.e., PhoD22 encoded by the AMG *phoD*22), respectively. Helices and sheets are colored in a rainbow scheme (from the N terminus in red to the C terminus in blue). Detailed information on individual computational protein models is provided in Supplementary Data 12. **b**, **f**, **j** show the activities of PPa1, PPa54, and PhoD22, respectively, with comparisons between each phage-encoded protein and its corresponsive controls. P1: the commercial pyrophosphatase (PPa, 1000 U mL$^{-1}$) of *Escherichia coli* was used as a positive control for both PPa1 and PPa54. N: the total protein from the recombinant *E.coli* cells transformed with an empty pET28a vector was used as a negative control for all three phage-encoded proteins. P2: the commercial recombinant *E.coli* alkaline phosphatase (1000 U mL$^{-1}$) was used as a positive control for PhoD22. The dots overlaying each bar represent the corresponding data points. **c**, **g**, **k** show the effects of pH on the activities of PPa1, PPa54, and PhoD22, respectively. **d**, **h**, **l** show the effects of temperature on the activities of PPa1, PPa54, and PhoD22, respectively. Data presented in (**b**–**d**, **f**–**h**, **j**–**l**) were mean values ± standard deviations from three independent experiments (i.e., *n* = 3). Relative activities of a given phage-encoded protein shown in individual panels (**c**, **d**, **g**, **h**, **k**, **l**) were calculated based on the highest activity of that protein reported within the corresponsive panel.

functionally assayed (Supplementary Fig. 21). The enzyme activity of PhoD22 was 62 U mL$^{-1}$, being much lower than that of the commercial *E.coli* PhoD (Fig. 3j). The optimal pH of the phage PhoD was 8.0 and its optimal temperature was 50 °C (Fig. 3k, e).

### Geographic distribution and community composition of P-acquisition phages and AMGs

Over 45% of the P-acquisition vOTUs (34/75) were detected in three or more habitat types (Supplementary Fig. 22). On average, the number of P-acquisition vOTUs detected in farmland was significantly higher than those of the other four habitat types (Supplementary Fig. 23a). Within each habitat type, individual ecosystems (sampling sites) differed greatly in the number of detectable P-acquisition vOTUs (Supplementary Fig. 23b–f and Supplementary Data 14). The number of P-acquisition vOTUs detected in soils was significantly correlated with soil total P (TP) or available P (AP) contents (Pearson's correlation coefficients: 0.25 and 0.47 respectively, *n* = 333, *P* = 2.2e−10 and 8e-6; Supplementary Fig. 24). As to the community composition of P-acquisition phages, the most predominant family in Gobi desert was *Mesyanzhinovviridae* (with an average relative abundance of 25%),

while the *Peduoviridae* family dominated most of ecosystems of the other four habitat types (with an average relative abundance ranging from 19 to 49%; Supplementary Fig. 23b–f).

The average number of P-acquisition AMGs detected in farmland was significantly higher than those of the other four habitat types (Fig. 4a). Within each habitat type, the number of detectable P-acquisition AMGs varied greatly among ecosystems (Fig. 4b–f). There were significant correlations between the number of P-acquisition AMGs detected in soils and the contents of soil TP and AP (Pearson's correlation coefficients: 0.17 and 0.39 respectively, *n* = 333, *P* = 1.3e−13 and 6.7e-4; Supplementary Fig. 24). When the relative abundances of the four categories of P-acquisition AMGs were taken into account, those involved in P transportation dominated most of the ecosystems of farmland, forest, and grassland, on average accounting for 40, 50, and 56% of the total gene abundance of P-acquisition AMGs in the three habitat types, respectively (Fig. 4b–d). In contrast, the predominant categories of P-acquisition AMGs in Gobi desert and mine wasteland were related to organic P mineralization, on average contributing 58 and 55% of the total gene abundance in the two habitat types, respectively (Fig. 4e, f).

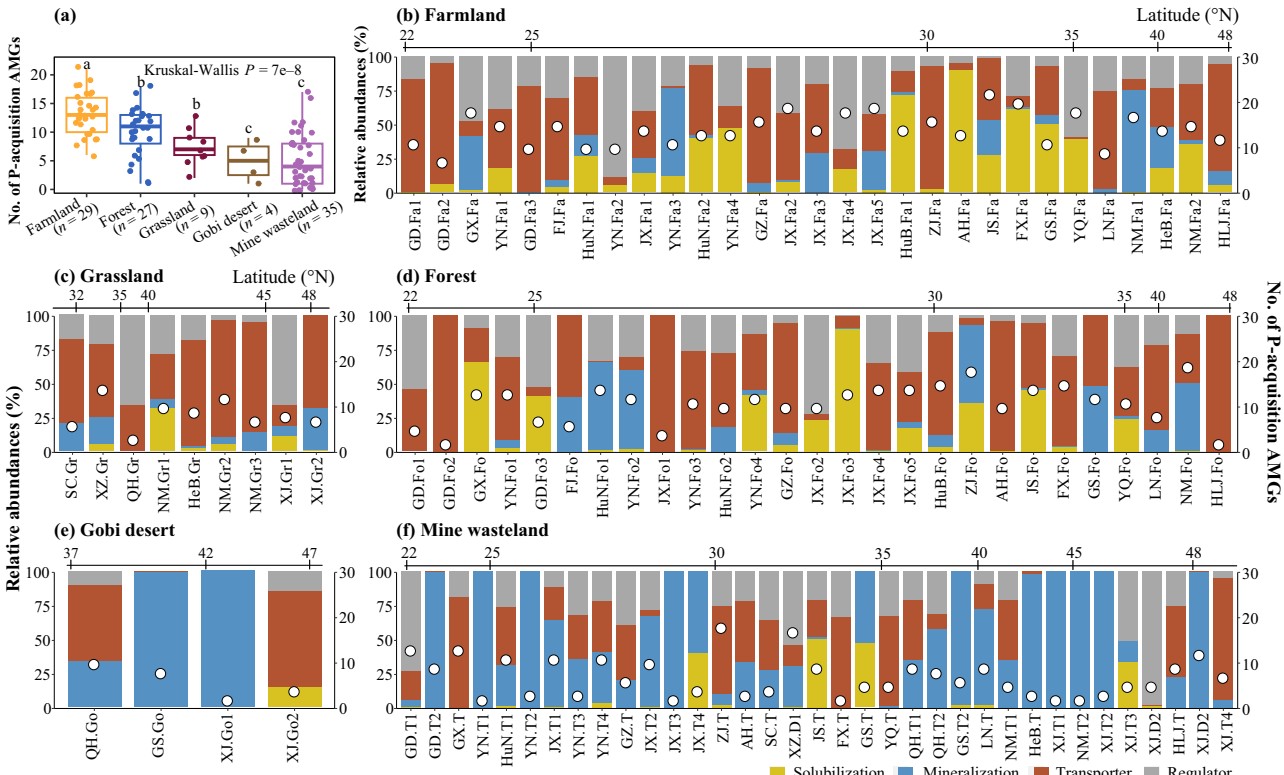

**Fig. 4 | The numbers and relative abundances of P-acquisition AMGs in individual sampling sites. a** The average numbers of all kinds of P-acquisition AMGs detected in the five habitat types. Horizontal lines represent the medians, while the boxes represent the interquartile ranges of the first and third quartiles. The vertical lines indicate the maximal and minimal values. The dots overlaying each bar represent the corresponding data points. Different letters on the top of the bars indicate significant differences between habitat types assessed with the two-sided Wilcoxon test, and the *P* value indicates the overall difference among all habitat types assessed with the Kruskal–Wallis test. **b**–**f** The relative abundances of the four categories of P-acquisition AMGs (illustrated with the bar charts, see scale values on the X-axis of the left-hand side of each panel) and the numbers of all kinds of P-acquisition AMGs (illustrated with white circles, see scale values on the X-axis of the right-hand side of each panel) detected in individual sampling sites. The four categories of P-acquisition AMGs (short for solubilization, mineralization, transporter, and regulator, respectively) are indicated by different colors. Sampling sites are first grouped as per their habitat types [Farmland (**b**), Forest (**c**), Grassland (**d**), Gobi desert (**e**), and Mine wasteland (**f**)], and then those within the same habitat type are arranged according to their latitudes (from south to north).

Thirty-six percent (27/75) of the P-acquisition vOTUs recovered from our soil metagenomes were detectable in a previously published global topsoil metagenome dataset[30] (Supplementary Data 15). As a whole, these 27 shared vOTUs were detected in up to 86% (248/288) of the metagenomes in the global dataset (Fig. 5a). With regard to the habitat origin of the shared vOTUs in our soil metagenomes, they were obtained from the forest (10 vOTUs), mine wasteland (9), and farmland (8), respectively (Supplementary Data 15). Despite this, the shared vOTUs were detected in all four habitat types of the global dataset (Fig. 5b and Supplementary Data 15): Arctic tundra (2 vOTUs), forest (26), grassland (14), and Mediterranean (8). While the number of the shared vOTUs detected in individual ecosystems of the global dataset varied greatly from one to 11 (Fig. 5a), a significantly positive correlation was observed between the number of the shared vOTUs detected in soils and soil TP content (Pearson's correlation coefficient: 0.17, $n = 260$, $P = 0.005$; Supplementary Fig. 25).

In our soil metagenomes, the 27 shared vOTUs encoded 13 kinds of P-acquisition AMGs (Supplementary Data 15). Remarkably, eight kinds of these AMGs (i.e., *phoR*, *phoU*, *pit*, *pstA*, *pstB*, *pstC*, *ugpE*, and *ugpQ*; Supplementary Data 16) encoded by six of the shared vOTUs, were also detected in 41 metagenomes of the global soil dataset, which spanned three habitat types (i.e., forest, grassland, and Mediterranean; Fig. 5c). Specifically, *pstA* and *pstB* were detected in three habitat types and the other six kinds of AMGs were detected only in forest (Fig. 5c). These results indicated that some soil phages could harbor the same P-acquisition AMGs regardless of the different terrestrial ecosystems.

### Relative influences of selected factors on soil P availability
Aggregated boosted tree (ABT) analysis was performed to compare the relative importance of microbial and environmental factors in determining soil P availability in different habitat types across China (Fig. 6; see Methods for details). When the five habitat types were considered together, TP was found to be the most important factor of soil P availability, followed in decreasing order by mean annual precipitation (MAT), relative abundance of prokaryotic P-acquisition genes in the soil metagenome (prokaryotic P-gene for short), relative abundance of phage P-acquisition genes in the soil metagenome (phage P-gene), and sites (Fig. 6a). When the five habitat types were considered separately, TP remained the most important factor in three habitat types (Fig. 6b, c, e). However, phage P-gene became more important than prokaryotic P-gene in farmland, Gobi desert, and mine wasteland (Fig. 6b, e, f). In detail, the relative influences of phage P-gene in the three habitat types were 18% (farmland), 18% (Gobi desert), and 19% (mine wasteland), correspondingly.

### Reconstruction of virus-host linkages
Applying a new integrated phage–host prediction method (iPHoP)[31], 35% (26/75) of the P-acquisition vOTUs identified in our metagenomes could be associated with a total of 28 hosts, which spanned ten bacterial and one archaeal phyla (Fig. 7a and Supplementary Data 17). Nine and seven of the predicted hosts were assigned to *Actinobacteriota* and *Proteobacteria*, respectively. Approximately 70% (18/26) of the vOTUs involved in host-virus linkages encoded *phoD*, *phoR*, and *ugpQ*, and they had a broad host range (Fig. 7a). Remarkably, vOTU44 that

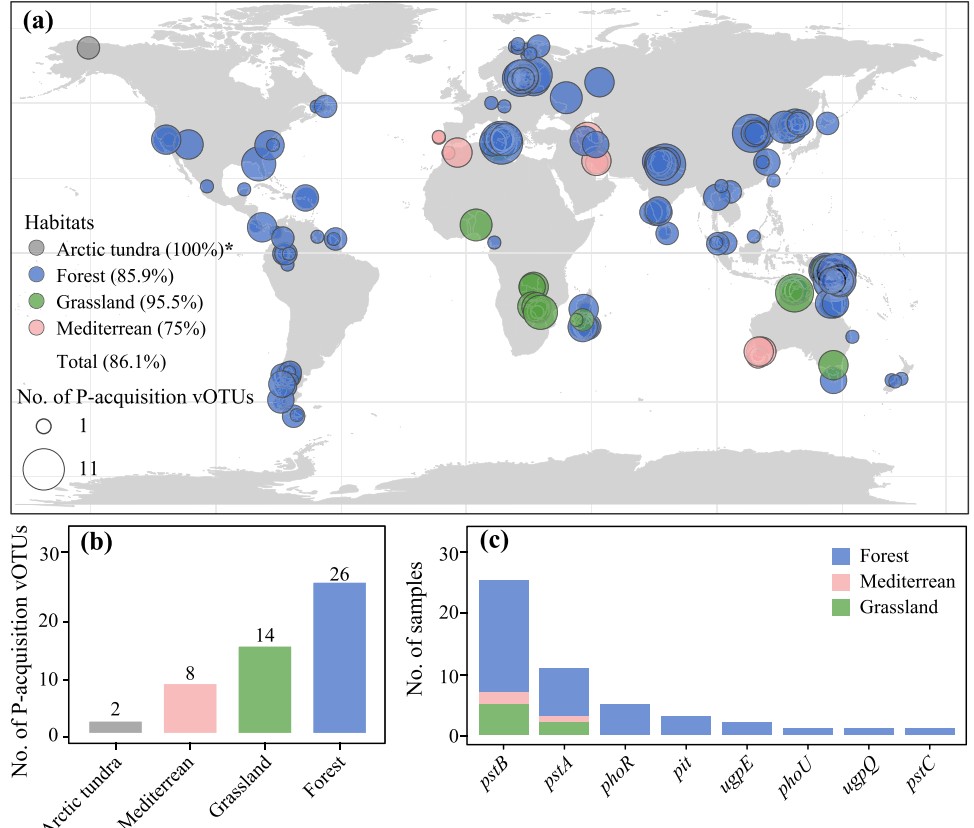

**Fig. 5 | Global distribution patterns of the P-acquisition vOTUs and AMGs identified in this study. a** Map showing the sampling sites of the global soil study (from a published global topsoil metagenome dataset)[30] and the numbers of P-acquisition vOTUs of our study that were also detected in individual sampling sites of the global soil study. Circles represent the sampling sites and are colored based on habitat types. Circle sizes reflect the numbers of P-acquisition vOTUs detected in the corresponding sampling sites. Circles at the same coordinates are stacked according to their sizes, with the largest one at the bottom. *, the value in the bracket following a given habitat type represents the percentage of samples with P-acquisition vOTUs in that habitat type. The world map was generated by the function map_ data ("world") in the R package ggplot2. **b** Histograms showing the total numbers of the P-acquisition vOTUs detected in individual habitat types of the global soil study. **c** Histograms showing the numbers of the global soil samples where the eight kinds of P-acquisition AMGs carried by the vOTUs shared by our study and the global soil study were detected. Habitat types are indicated by different colors. Detailed information is provided in Supplementary Data 15, 16.

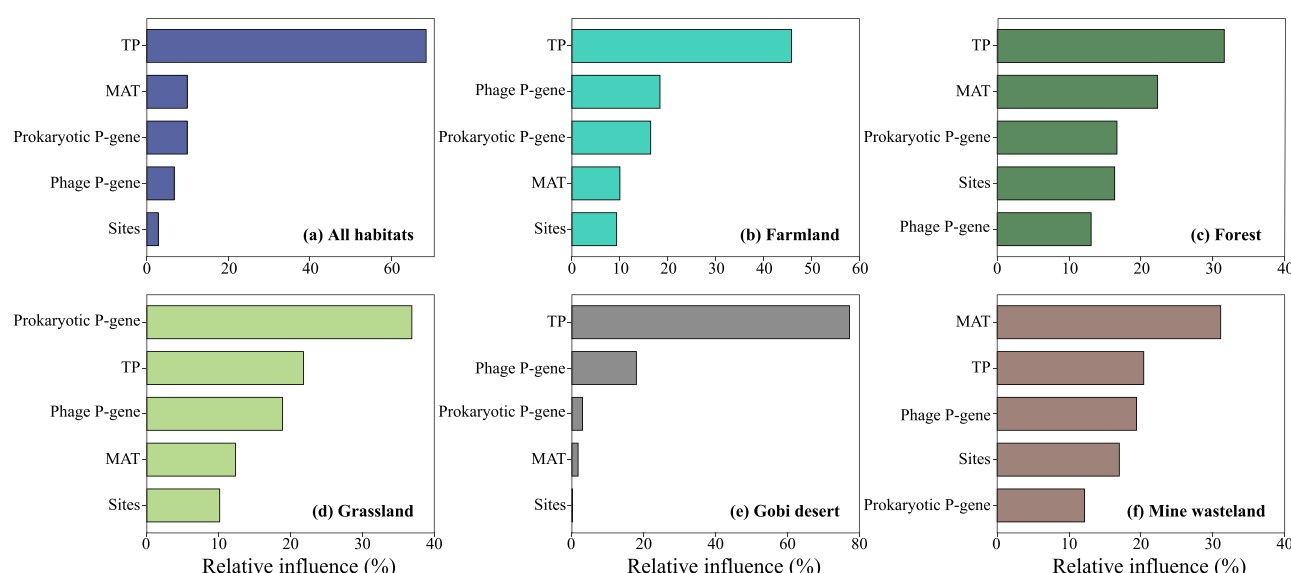

**Fig. 6 | Relative influences of selected factors on soil P availability evaluated by aggregated boosted tree (ABT) models. a** ABT results for the five habitat types as a whole. **b**–**f** ABT results for farmland, forest, grassland, Gobi desert, and mine wasteland, respectively. TP total phosphorous, MAT mean annual precipitation, Prokaryotic P-gene relative abundance of prokaryotic P-acquisition genes in the metagenome, Phage P-gene relative abundance of phage P-acquisition genes in the metagenome.

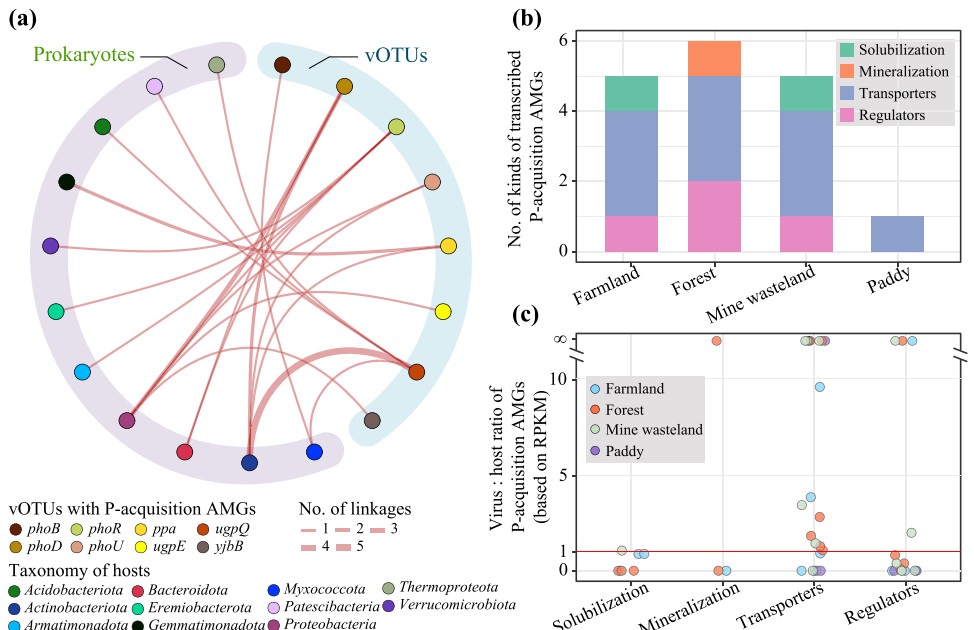

**Fig. 7 | P-acquisition vOTU-host linkages and gene transcription profiles of P-acquisition AMGs. a** Pairwise P-acquisition vOTU-host linkages identified in this study. vOTUs are shown on the right-hand side of the panel and colored according to the kinds of P-acquisition AMGs that they encoded, while the hosts are shown on the left-hand side of the panel and colored according to their taxonomy. The number of linkages between individual vOTUs and their hosts are proportional to the sizes of those lines linking them. Detailed information of the P-acquisition vOTU-host linkages is provided in Supplementary Data 17. **b** Histograms showing the numbers of kinds of P-acquisition AMGs transcribed in individual habitat types.

The numbers of the transcribed genes belonging to the four categories of P-acquisition AMGs in each habitat type are also indicated by different colors. **c** The transcript ratios of phage:host gene pairs related to the four categories of P-acquisition AMGs in individual samples. Each circle in the panel represents a sample and is colored according to its habitat type. RPKM, reads per kilobase per million mapped reads. Detailed information on the public metatranscriptomes used to generate the results in (**b**) and (**c**) was provided in Supplementary Data 18, while the numbers of the samples where these P-acquisition AMGs were detected are shown in Supplementary Data 19.

encoded *ugpQ* was associated with hosts from both the Bacteria (*Actinobacteriota*) and Archaea (*Thermoproteota*) domains (Supplementary Data 17). Regardless of their host and habitat type, most of the UgpQ proteins encoded by the viral *ugpQ* genes recovered from our soil metagenomes clustered together in the phylogenetic tree (Supplementary Fig. 26a). A similar pattern was observed for the phage-associated PhoD proteins (Supplementary Fig. 26b). In contrast, those proteins encoded by other P-acquisition AMG kinds (i.e., PPa, PstA, PstB, PstC, PstS, PhoB, PhoR, and PhoU) tended to cluster separately with corresponding reference proteins of different bacterial phyla (Supplementary Figs. 20, 27, 28).

## Transcription profiles of P-acquisition AMGs

To further assess the potential contribution of P-acquisition phages to the biogeochemical cycling of P in soils, the gene transcription profiles of the P-acquisition AMGs identified in our soil metagenomes were explored with 32 public soil metatranscriptomes of farmland, forest, mine wasteland, and paddy from China (eight metatranscriptomes were selected for each habitat type, Supplementary Data 18). A total of seven AMG kinds (i.e., *ppa*, *phoD*, *pstA*, *pstB*, *pstC*, *phoU*, and *phoR*) involved in the four primary processes of microbial P-acquisition were detected in these metatranscriptomes (Supplementary Data 19 and Fig. 7b). Among them, *pstB* genes were transcribed in all four habitat types, and those of three other AMG kinds (i.e., *pstA*, *pstC*, and *phoU*) were transcribed in three habitat types. The transcripts of *phoD* and *phoR* genes were detected only in the forest. Taken together, five, six, five, and one kind of AMGs were found to be transcribed in farmland, forest, mine wasteland, and paddy, respectively (Fig. 7b). As to the phage:host P-acquisition gene pairs identified in our soil metagenomes (see Methods for details), the ratios of their phage to host transcripts in the public

metatranscriptomes differed greatly depending on not only gene kind but also habitat type (Fig. 7c). For instance, almost all the transcript ratios of phage:host gene pairs associated with inorganic P solubilization were lower than one, while those of P transportation in mine wasteland varied from 0 to infinity (i.e., phage gene transcript abundance >0, but host gene transcript abundance = 0). Remarkably, over 45% (22/48) of the transcript ratios of phage:host gene pairs recorded in this study were greater than one (Fig. 7c), indicating that under certain circumstances, phages likely play a more important role than their host in the P biogeochemical cycle.

## A schematic model of phage auxiliary metabolism in bacterial P-acquisition

Based on our metatranscriptomic results, we proposed a schematic model of how terrestrial P-acquisition phages can impact the P-acquisition processes of their hosts (Fig. 8). In low-phosphate soil environments, the following three scenarios likely occur: (i) a phage-encoded PPa is expressed in the host cell and released into the environment, wherein it catalyzes the hydrolysis of pyrophosphate into phosphate; (ii) a phage-encoded PhoD is expressed and released into the environment, wherein it catalyzes the hydrolysis of phosphomonoester into phosphate; and (iii) a phage-encoded PhoR is expressed and phosphorylates a PhoB of the host, resulting in the expression of phage-encoded PstSCAB transport system (and also that of the host). Remarkably, an elevated expression of the high-affinity phosphate transport system is beneficial for the host to import more phosphate from the environment into its cytoplasm. When the cytoplasmic phosphate level exceeds a certain threshold that raises toxicity to the host cell, the fourth scenario likely occurs: (iv) a phage-encoded PhoU is expressed and inhibits the role of PhoR on PhoB, thus suppressing the expression of PstSCAB transport system.

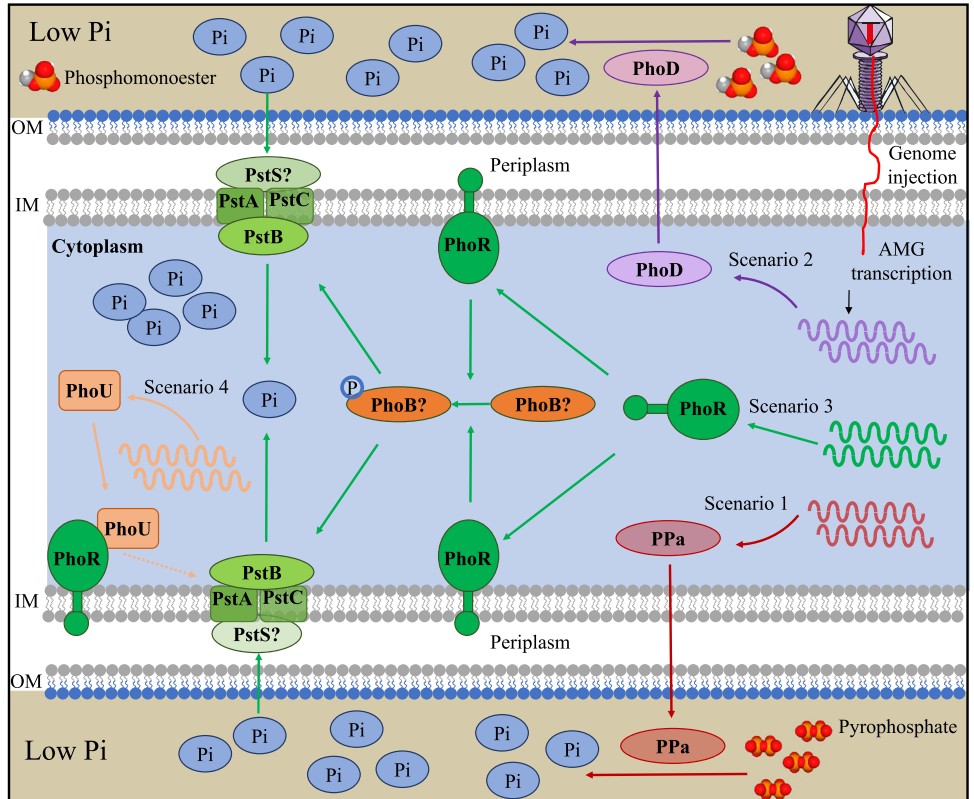

**Fig. 8 | A schematic model of phage auxiliary metabolism in bacterial P-acquisition.** In low-P terrestrial ecosystems, P-acquisition phages can impact their hosts' P-acquisition processes via four scenarios, which are indicated by red (Scenario 1), purple (Scenario 2), green (Scenario 3), and orange (Scenario 4) arrows, respectively. Positive and negative regulations of P-acquisition AMGs are indicated by solid and dashed arrows, respectively. Seven proteins (i.e., PPa, PhoD, PhoR, PhoU, PstA, PstB, and PstC) encoded by the P-acquisition AMGs detected in the investigated metatranscriptomes (listed in Supplementary Data 18) are shown; whilst two additional proteins (i.e., PhoB and PstS) are shown and marked by a question mark, as the transcripts of phage *phoB* and *pstS* were not detected in the metatranscriptomes. The PhoB with a capital letter 'P' indicates that it was phosphorylated. The substrates of PPa (i.e., pyrophosphate) and PhoD (i.e., phosphomonoester) are shown, respectively. For simplicity, only the outer and inner cell membranes for Gram-negative bacteria are shown. IM inner membrane, OM outer membrane, Pi phosphate.

## Discussion

There has been extensive interest in the impact of viral auxiliary metabolism on the global biogeochemical cycling of sulfur and nitrogen[32,33]. However, viral auxiliary metabolism associated with microbial P-acquisition from the environment has been poorly explored. To the best of our knowledge, 11 kinds of P-acquisition AMGs have been characterized in aquatic habitats so far, including oceans, estuaries, and acid mine drainage sediments[18,19,34–37] (see Supplementary Data 9 for more details). As to terrestrial habitats, only four kinds of P-acquisition AMGs have been reported in industrial or agricultural soils[21–23]. Through a large-scale metagenomic survey, we recovered a total of 17 kinds of P-acquisition AMGs (Fig. 2a). Among them, 11 kinds (i.e., *ppx*, *phnP*, *phnW*, *ugpQ*, *pit*, *pstA*, *pstB*, *pstC*, *ugpE*, *phoR*, and *phoU*) were not reported previously. This finding greatly expands the diversity of P-acquisition AMGs. The seemingly saturated accumulative curve of P-acquisition vOTUs with the sample size of our soil metagenomes (Fig. 1b) can be at least partly explained by a low recovery rate in viral sequences from the metagenomes. Thus, a deeper metagenomic sequencing effort and/or an application of metaviromics will probably capture more novel kinds of AMGs encoded by P-acquisition phages. Among the 11 kinds of P-acquisition AMGs identified in public databases (Supplementary Data 10), seven (i.e., *ugpQ*, *pstA*, *pstB*, *pstC*, *ugpB*, *phoR*, and *phoU*) were not reported in previous studies, suggesting that public databases should not be overlooked in future studies on viral auxiliary metabolism.

The identification of diverse kinds of P-acquisition AMGs in our study indicates several major metabolic strategies that may be employed by terrestrial phages to assist their hosts to cope with P-limited conditions. One such strategy is the *phoD*-associated alkaline phosphatase pathway, wherein bioaccessible phosphate is released from recalcitrant phosphomonoesters[10]. The work of Zheng et al. detected three phage *phoD* genes in soils, but that study did not discuss them[21]. The number of phage *phoD* genes recovered from our soil metagenomes was up to 23, being much higher than those of the other 16 kinds of P-acquisition AMGs (Fig. 2a). Moreover, *phoD* was the only one AMG kind that can be found in all five habitat types (Fig. 2a). Coincidentally, 16 *phoD* genes were also identified in public viral genomes, among which 11 were originated from soils (Supplementary Data 10). Another major strategy is to acquire bioaccessible phosphate by PPa proteins, which can catalyze the hydrolysis of pyrophosphate into phosphate[38]. Eight vOTUs recovered from soils of three habitat types across China (Fig. 2 and Supplementary Data 7), and one public *Caudoviricetes* genome (DTR_349854) from rhizosphere soil (Supplementary Data 10) were found to encode *ppa*. There are two possible reasons for the occurrence of diverse phage *phoD* and *ppa* genes in soil. First, as a primary organic P form, phosphomonoester can account for up to 70% of total organic P in soil[39]. Second, pyrophosphate comprises a relatively large portion of the inorganic P pool in soil[40]. The functional validation of representative phage-encoded *phoD* and *ppa* genes (Fig. 3) suggests that the diversity of the related genes (or, more exactly, that of their associated proteins) is beneficial

for bacteria to acquire phosphate from a variety of P-limited environments. This notion is further supported by the observation that some phage *phoD* and *ppa* genes were transcribed in soils of forest, farmland, and mine wasteland (Fig. 7b, c).

A third major strategy is relevant to the Pst high-affinity phosphate-specific transport system. While several previous studies had documented the *pstS* genes encoded by aquatic phages[12,14,18,41], those genes (i.e., *pstA*, *pstB*, and *pstC*) encoding the other three components of the Pst system have not yet been identified in any phages. In this study, four vOTUs recovered from our soil metagenomes and one public phage genome were found to encode a complete Pst system, respectively (Supplementary Data 7, 10). Moreover, the transcripts of phage *pstA*, *pstB*, and *pstC* genes were detected in three or more habitat types, respectively (Fig. 7b, c). In aquatic cyanobacterial phages, the transcription of *pstS* was reported to be regulated by the host's phoR-phoB system[12,41]. However, our results allow us to propose a new scenario: certain phages have the potential to regulate host phosphate uptake machinery to obtain extracellular P. On the one hand, 13 vOTUs and one public phage genome (Supplementary Data 7 and 10) were revealed to carry *phoR* that encodes the activator of the response regulator PhoB[13]. On the other hand, *phoU* encoding a negative regulator in phosphate signaling[15], was identified in seven vOTUs and one public phage genome (Supplementary Data 7, 10). Additionally, the transcripts of phage *phoR* and *phoU* genes were detected in some investigated metatranscriptomes (Fig. 7b, c). Similar to the proposed scenario, certain phages were demonstrated to have the ability to regulate nitrogen fixation and nitrate respiration in their hosts by harboring *nifL* and *narL* genes[42,43].

The acquisition of P by infected host cells and its utilization for both host growth and phage replication are significant for phage–host evolution[33]. Phage particles are generally enriched in P compared with the baseline elemental stoichiometry of the host cells, implying that phages must concentrate P to reproduce[44]. Phages can recycle intracellular nucleotides or use extracellular P resources for self-replication, which depends largely on P availability in the environment[45,46]. The widespread distribution of various P-acquisition AMGs in soils of different habitat types (Figs. 5, 7) hints that terrestrial phages prefer to exploit extracellular P to alleviate the resource bottleneck during infection. A similar phenomenon has been observed in P-limited oceans, wherein cyanobacterial host cells are believed to have a reduced intracellular content of P[46], presumably forcing phages to rely more heavily on extracellular P to replicate[16,33]. In this way, chronic P scarcity likely has an important impact on viral evolution as well as that of their hosts. The possibility seems to be supported by our finding that the calculated $d$N/$d$S values for P-acquisition AMGs were <1 (Supplementary Data 13). Such values suggest that the related phages in terrestrial habitats are under selective pressure to keep their P-acquisition AMGs functional, as otherwise, deleterious mutations will result in the loss of these AMGs[32]. Although further evidence is needed to validate the putative effect of P scarcity on viral evolution, the implications of this evolutionary scenario can be broad for at least two reasons. First, the relevant P-acquisition AMGs were identified in multiple taxonomic lineages of viruses (Supplementary Data 8). Second, these viruses likely interact with various hosts from different phyla or even different domains (Fig. 7a).

Due to their ability to reprogram host cell metabolism, the biogeochemical influence of viruses in nutrient element cycling has been known to begin upon infection[14]. Moreover, it is becoming increasingly clear that viral infection has the potential to be a major contributor to biogeochemical processes at regional and even global scales, well beyond the metabolism of individual host cells[33]. In agreement with this notion, we obtained two lines of evidence indicating the broad biogeochemical impacts of P-acquisition phages. On the one hand, our ABT analysis showed that phage P-gene had an important effect on soil P availability at the country scale, especially for farmland, Gobi desert,

and mine wasteland (Fig. 6). On the other hand, in over 45% cases examined in this study, the transcription levels of focal P-acquisition AMGs (e.g., *pstS*, *pstA*, *pstB*, and *pstC*) were higher than those of their corresponding host genes (Fig. 7c).

In summary, we reveal the hidden diversity and widespread distribution of terrestrial P-acquisition phages and AMGs, verify the enzymatic activities of three P-acquisition AMGs encoded by terrestrial phages, and provide metatranscriptomic evidence for the important ecological roles of P-acquisition phages in terrestrial habitats. Our results highlight that future research on biogeochemical P cycling should incorporate the roles of phages.

## Methods

### A country-scale sample collection and physicochemical analysis
A total of 333 soil samples representing five different types of terrestrial habitat (29 farmlands, 27 forests, 4 Gobi deserts, 9 grasslands, and 42 mine wastelands; i.e., a total of 111 ecosystems) across 22 provinces in China, were collected between July and August 2018 (Fig. 1a and Supplementary Data 1). The geographical coordinate of each ecosystem was recorded using the Global Positioning System while sampling, which was used to attain the MAT of each ecosystem from WorldClim version 2 (www.worldclim.org). At each ecosystem, three soil samples were collected at a depth of 0–20 cm according to the method described previously[47], and transported back to the laboratory on ice. Each soil sample was divided into two parts, with one part being air-dried, ball-milled, sieved, and homogenized for physicochemical analyses, while the other being stored at −80 °C for DNA extraction. Bioavailable soil P was determined according to the sodium bicarbonate (Olsen) method, and total soil P was determined using the molybdate blue colorimetric method.

### DNA extraction and metagenomic sequencing
Soil DNA was extracted using the FastDNA Spin kit (MP Biomedicals, USA) following the manufacturer's protocol. DNA quality was assessed with the NanoDrop 2000 spectrophotometer (Thermo Scientific, USA). Each high-quality (HQ) soil DNA sample was further purified and used to construct a sequencing library (~ 300 bp average insert size) for whole metagenome sequencing. Subsequently, it was shotgun-sequenced on the Illumina HiSeq 2500 platform with PE150 mode (Illumina, USA).

### Viral sequence identification and dereplication
Metagenomic reads were filtered by quality using in-home Perl scripts, which included eliminating duplicated reads, removing reads with ≥5 "N", and filtering low-quality reads (quality score ≥30)[9]. HQ reads were then assembled into contigs using MEGAHIT (version 1.2.9) with the parameters "k-min 35, k-max 95, and k-step 20"[48]. Contigs longer than 10 kb were screened by VirSorter2 (v2.2.3; default parameters)[49] and were further refined by identifying and removing potential host contaminants using CheckV (v0.9.0; default parameters)[50]. The predicted viral sequences were clustered into vOTUs following standard guidelines at 95% identity and 85% alignment fraction using dRep (v3.3.0)[51]. The longest sequence in each cluster was selected as the representative for that cluster.

### Phage P-acquisition gene identification and vOTU validation
For each vOTU, ORFs were predicted using Prodigal (v2.6.3; -p meta)[52], and the resulting protein sequences were compared against the KEGG databases to obtain their functional annotation using Diamond (v0.9.24.125) BLASTp (*e*-value $10^{-5}$, coverage 50%, and identity 50%) and HMMER 3.3.2 (gene-specific scores and score types provided at ftp://ftp.genome.jp/pub/db/kofam/ko_list.gz)[53,54]. All investigated P-acquisition genes with their KO numbers were listed in Supplementary Data 2 and the respective cut-off scores used for P-acquisition gene search by HMMER were listed in Supplementary Data 3.

The vOTUs carrying P-acquisition genes were further validated as phages using three methods: (i) VIBRANT (v1.2.1; default parameters)[55]; (ii) SOP used in Sullivan Lab for viral identification based on VirSorter2 and CheckV (dx.doi.org/10.17504/protocols.io.bwm5pc86); and (iii) manual curation based on VIBRANT annotations (i.e., the KEGG, Pfam, and VOG databases) with three established criteria[56]. The criteria were described as follows: (i) scaffolds with at least five hits to viral protein families, the number of genes with KO term assignments is <20%, Pfam-assigned genes <40%, and the total number of genes with viral protein families hits >10%; (ii) scaffolds with at least five hits to viral protein families and the number of genes with hits to viral protein families is greater than the number of genes with hits to Pfams; and (iii) scaffolds with at least five hits to viral protein families and the number of genes with hits to viral protein families is at least 60%. Only the vOTUs meeting one of the above criteria were considered as viruses. Detailed information on the resultant vOTUs was provided in Supplementary Data 4.

To validate the potential P-acquisition AMGs of the resultant vOTUs, DRAM-v in DRAM (v1.2.0) was used. According to DRAM-v, the genes with auxiliary scores of 1–3 and AMG flag of −M and/or −F were considered as AMGs[57]. Eight genes with auxiliary scores of 1–3 and AMG flag of −T were also considered as AMGs, because the vOTUs harboring these genes were validated as phages by the above-mentioned three methods. As per DRAM-v, a gene is not considered an AMG if it is in a row of three metabolic genes (with an AMG flag of −B and an auxiliary score of 4). However, the genome of one isolated bacteriophage, *Campylobacter* phage A18a (GCA_002956935.1), harbored *phoU* and *pstBACS* in a row, suggesting that *pst* gene cluster can actually exist in a phage genome. Therefore, the *pstSCAB* gene clusters (with auxiliary scores of 4) present in two vOTUs (i.e., vOTU47 and vOTU60) were considered AMGs and included in our subsequent analyses. The genomic contexts of the P-acquisition AMGs recovered from our soil metagenomes based on VIBRANT and DRAM-v annotations were provided in Supplementary Data 5 and 6. The validated AMGs were then dereplicated using CD-HIT (v4.8.1) based on 95% sequence similarity and 90% alignment coverage[58].

## Taxonomic assignment of P-acquisition vOTUs

GeNomad (v1.7.0; default parameters)[24] was used for the taxonomic classification of the P-acquisition vOTUs. However, only a few vOTUs could be assigned to specific viral families or genera. Thus, the family-level and finer classifications of the P-acquisition vOTUs were alternatively performed by PhaGCN (v2.0; default parameters)[25] using a recommended cut-off score >0.5. All taxonomic assignments were subjected to manual inspection.

## Identification of P-acquisition phages from public databases

A total of 68,904 viral sequences from the IMG/VR and NCBI GenBank databases (February 2022) were downloaded. The viral sequences longer than 10 kb were used for protein sequence prediction by Prodigal (-p meta). The protein sequences were compared (e-value $10^{-5}$, coverage 50%, and identity 50%) against the KEGG databases using Diamond (v0.9.24.125) BLASTp (e-value $10^{-5}$, coverage 50%, and identity 50%) and HMMER 3.3.2 (gene-specific scores and score types) as described above. The P-acquisition AMGs of three representative viral genomes from the public datasets and their genomic contexts were provided in Supplementary Data 10, 11.

## Phylogenetic analysis of P-acquisition AMGs from our metagenomes

Protein sequences of the focal P-acquisition genes from reference prokaryotes were downloaded from the UniProt database (accessed July 2022). The reference sequences that have been reviewed or published were selected, and then manually filtered for accurate annotations. These sequences were clustered by 70% sequence similarity using CD-HIT (v4.8.1), and the representative sequences from individual clusters were aligned with the corresponding AMG protein sequences using MAFFT (v7.490, default settings). Alignments were subjected to maximum likelihood phylogenetic tree reconstruction using IQ-TREE (v1.6.12, −bb 1000 −alrt 1000 −m MFP)[59]. Trees were visualized using the Interactive Tree of Life online interface[60].

## Sequence alignment of P-acquisition AMGs and protein structure modeling

The protein sequences from each P-acquisition AMG family identified in our metagenomes were aligned along with their reference sequences respectively, using ClustalW with slow/accurate setting parameters (https://www.genome.jp/tools-bin/clustalw). The alignments were manually corrected and later visualized by ESPript 3.0[61]. One representative protein sequence from each AMG family was structurally modeled using Phyre2[62] in normal modeling mode to confirm and further resolve functional predictions (Supplementary Data 12).

## Calculation of dN/dS for P-acquisition AMGs

To calculate the $d$N/$d$S ratios between gene pairs of each P-acquisition AMG kind, dRep (v3.3.0) was used to compare protein sequences of ten P-acquisition AMG kinds: *phoD* ($n = 24$), *ugpQ* ($n = 21$), *phoR* ($n = 13$), *phoB* ($n = 11$), *ppa* ($n = 8$), *phoU* ($n = 7$), *pstS* ($n = 6$), *pstA* ($n = 4$), *pstB* ($n = 4$), and *pstC* ($n = 4$), separately (dRep compare−SkipMash−S_algorithm goANI). The other seven P-acquisition AMG kinds were excluded from the calculation because there was only one sequence for each of them. A published auxiliary script (dnds_from_drep.py) was used to calculate $d$N/$d$S ratios based on the dRep outputs of various AMG pairs[63]. Specifically, protein sequences of each AMG kind were clustered individually at 70% identity over at least 70% of their lengths. Sequences in each cluster were separated into all possible pairs and each pair was subjected to the $d$N/$d$S calculation. The focal AMG pairs and their respective $d$N/$d$S values were listed in Supplementary Data 13.

## Experimental verification of phage ppa and phoD genes

The gene sequences of three representative P-acquisition AMGs (i.e., *ppa*1, *ppa*54, and *phoD*22) were optimized for *E. coli*, synthesized, and cloned into the pET28a plasmid, respectively[38]. The three kinds of transgenic plasmids were each separately transformed into *E. coli* BL21 cells. These recombinant cells were incubated overnight in lysogeny broth with 0.17 mM kanamycin at 37 °C on a rotary shaker (220 r.p.m.). For each of the three kinds of overnight cultures, 10 mL was used to inoculate 1000 mL of fresh lysogeny broth. After the cells grew to an optical density of 0.6 at 600 nm, target protein expression was induced by the addition of isopropyl-β-ᴅ-thiogalactoside at a dose of 0.5 mM. The cultures were incubated at 20 °C and shaken at 200 r.p.m. for 12 h. Subsequently, cells in the cultures were harvested by centrifugation (5000 × $g$ for 10 min).

The harvested cells were resuspended in 50 mL of cell breakdown buffer containing 0.05 M Tris-HCl and 10% glycerol, and then sonicated for 36 × 15 s (20% amplitude, 5 s + 10 s pulses) using the Ultrasonic Homogenizer JY92-IIN (Scientz, China). The cell lysate was obtained by centrifugation (14,000×$g$ for 10 min) and purified using the His Ni-NTA Agarose Resin (Yeasen Biotechnology, China) according to the manufacturer's instructions. Purification fractions were examined by SDS−PAGE, and protein concentrations were determined by the NanoDrop 2000 spectrophotometer (Thermo Scientific, USA) at 280 nm.

PPa activity was measured by determining the amount of inorganic phosphate (Pi) released during enzymatic hydrolysis of $Na_4P_2O_7$. To this end, 200 μL of reaction mixture containing 2 mM $Na_4P_2O_7$, 2 mM $MgCl_2$, 100 mM Tris-HCl (pH 7.5), and diluted PPa (0.001 μM for PPa1 and 11.5 μM for PPa54) was incubated at 25 °C for 10 min. Reactions were stopped by adding malachite green reagent into the

mixtures (2:1 in volume). The Pi concentrations in the mixtures were determined at 620 nm using a spectrophotometer (Shimadzu, Japan). To assess the temperature and pH dependency of the activities of the two PPa enzymes, individual assay conditions were altered when necessary. An *E.coli* PPa (Yeasen Biotechnology, China) was used as a positive control, and the recombinant *E.coli* BL21 cell lysate with an empty pET28a vector was used as a negative control. The amount of enzyme required for catalytic hydrolysis of $Na_4P_2O_7$ to produce 1 μmol Pi per minute was defined as 1 unit of enzyme activity.

Alkaline phosphatase (PhoD22) activity was measured by determining the amount of para-nitrophenol (*p*NP) released during enzymatic hydrolysis of *p*-nitrophenyl phosphate (*p*NPP). For this purpose, an alkaline phosphatase assay kit (Beyotime Biotechnology, China) was used. Two hundred μL of the reaction mixture with diluted alkaline phosphatase (11.4 μM) was incubated at 37 °C for 10 min. The reaction was stopped by adding the reaction termination solution provided in the kit into the mixture (1:1 in volume). The *p*NP concentration in the mixture was determined at 405 nm using a spectrophotometer (Shimadzu, Japan). To assess the temperature and pH dependency of the enzymatic activity, individual assay conditions were altered when necessary. An *E.coli* alkaline phosphatase (Sangon Biotechnology, China) was used as a positive control, and the recombinant *E.coli* BL21 cell lysate with an empty pET28a vector was used as a negative control. The amount of enzyme required for catalytic hydrolysis of *p*NPP to produce 1 μmol *p*NP per minute was defined as 1 unit of enzyme activity.

### Metagenomic mapping and relative abundance calculation

HQ metagenomic reads from each sample (soil metagenome) was mapped against the dereplicated P-acquisition vOTU sequences of our study by Bowtie2 (v2.3.4.1) with "--very sensitive" mode[64]. The coverage of each P-acquisition vOTU in a given sample was calculated as its scaffold coverage in that sample, weighed by its length in base pairs. The normalized coverage of each P-acquisition vOTU in a given sample was calculated as its coverage divided by the number of reads in that sample, and then multiplied by the mean value of the number of reads in all samples[65]. Relative abundance profiles of individual P-acquisition vOTUs in a given sample were generated by transferring the normalized coverage table to a proportional table (sum to 100% within each sample). The number of P-acquisition vOTUs in each sample was counted based on the relative abundance table shown in Supplementary Data 14. The community composition of P-acquisition vOTUs in a given sample was analyzed with the relative abundances of them in that sample according to their taxonomy. As proposed previously[18,34], to avoid the potential effects of host-derived reads on the calculation of relative abundances of the P-acquisition AMGs, the relative abundances of P-acquisition vOTUs were used to represent their AMGs' relative abundances. The Kruskal–Wallis test was used to test for the overall difference across all habitat types in the number of P-acquisition vOTUs or AMGs. Only if the Kruskal–Wallis statistic is statistically significant, pairwise comparisons between habitat types were performed using the Wilcoxon rank-sum test with Bonferroni-Holm correction.

### Mapping the P-acquisition vOTUs to a global topsoil metagenome dataset

To project the distribution of the P-acquisition vOTUs identified in this study onto a global scale, a total of 288 global topsoil metagenomes[30] were downloaded from the European Bioinformatics Institute Sequence Read Archive database (PRJEB24121). Raw reads were processed as described above, and then reads shorter than 100 bp were removed. HQ reads from each metagenome was separately mapped to each P-acquisition vOTU sequence recovered in this study using Bowtie2 (--very sensitive). The mapped vOTUs were further manually scrutinized to determine whether the P-acquisition AMGs on these vOTUs were also mapped by the reads from the same metagenome.

### Relative abundance calculation of prokaryotic P-acquisition genes

The metagenomic assemblies (≥500 bp and viral sequences excluded) of our soil samples were used for protein sequence prediction by Prodigal (v2.6.3; -p meta). The protein sequences were compared against the KEGG databases as described above, and those involved in microbial P-acquisition were identified by KEGG annotation hits were selected. The P-acquisition genes were first dereplicated, and then mapped by HQ reads using the same methods as described above. The normalized gene coverage table was generated as mentioned above.

### ABT analysis

ABT analysis is a machine learning method based on a decision tree, which aims to quantitatively evaluate the relative importance of certain explanatory variables in explaining the variation of a response variable of interest[66]. A major reason why we use ABT analysis here was that its prediction accuracy is generally higher than those of other methods such as random forest, bagged trees, and generalized additive models[66]. To explore the relative influence of environmental and microbial variables on soil P availability, ABT analysis was carried out using the gbm package within the R statistical computing environment. According to a previous study[47], TP was the most important predictor for soil P availability among edaphic variables, and MAT was the most important predictor among climatic variables. We therefore chose TP and MAT as environmental variables in our ABT analysis. As to microbial variables, phage P-gene (the focus of this study) and prokaryotic P-gene were selected. The "sites" was used in ABT analysis as a random variable to reflect the potential influences of the three soil samples collected from each ecosystem.

### Host prediction

To depict the virus-host links of the P-acquisition vOTUs, iPHoP (v1.3.3) was used for host prediction[31]. Both the default iPHoP database and custom metagenome-assemble genomes (MAGs) from our 333 soil metagenomes were used to maximize host prediction. Briefly, MAGs were generated from our metagenomes using metaBAT2 (v2.12.1; default parameters)[67], and the obtained MAGs were then evaluated by CheckM (v1.2.0)[68]. The MAGs with completeness >50% and contamination <10% were remained for host prediction. Taxonomic annotation for these MAGs was performed using GTDB-Tk (v2.1.1)[69].

### Metatrascriptomic mapping

A total of 32 soil metatranscriptomes of four terrestrial habitat types (i.e., farmland, forest, mine wasteland, and paddy) were obtained from two previous studies (Supplementary Data 17)[70,71]. Raw reads of the selected metatranscriptomes were filtered by fastp (v0.23.2) with the parameters --cut_mean_quality 20 and -l 50[72]. The rRNA sequences from prokaryotes and eukaryotes were removed by SortMeRNA (v4.3.6) with default parameters[73]. As described previously[32], the phage–host gene pairs for P-acquisition genes recovered from our metagenomes were identified by constructing phylogenetic trees containing individual kinds of P-acquisition AMGs and their bacterial counterpart gene encoding proteins recovered from the same metagenome, wherein the nearest relative of a given AMG was thought to be originated from the host of the phage carrying that AMG. Subsequently, the filtered metatranscriptome reads were mapped to the abovementioned phage–host gene pairs using Bowtie2 (--very-sensitive), respectively. The transcription level of each gene in each metatranscriptome was normalized to reads per kilobase per million mapped reads (RPKM) value.

## Reporting summary

Further information on research design is available in the Nature Portfolio Reporting Summary linked to this article.

## Data availability

Metagenomic sequencing data generated in this study have been deposited in the NCBI BioProject database under the accession number PRJNA1085405. The vOTU sequences of this study have been deposited in the ENA Sequence Read Archive database under the accession number PRJEB60228. Previously published 288 global topsoil metagenomes are available in the ENA Sequence Read Archive database under the accession number PRJEB24121. Previously published 32 metatranscriptomes are available in ENA and NCBI Sequence Read Archive databases (accession nos. PRJNA716119, PRJNA1056670, and PRJEB42658), and the specific accession numbers for individual samples are listed in Supplementary Data 18. Datasets used and/or analyzed in the study can be found in Supplementary Data 1–19.

## Code availability

The codes used in this study are available on GitHub[74].

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

## Acknowledgements

We thank Professor AJM Baker (Universities of Melbourne and Queensland, Australia, and Sheffield, UK) for his help in English language editing. This work was supported financially by the National Natural Science Foundation of China (Grant nos. 41622106 and 42077117 to J.-t.L., and no. 42177009 to J.-L. Liang) and the National Key Research and Development Program of China (no. 2023YFC3207300 to J.-t.L.).

## Author contributions

J.-t.L., J.-L.Liang, and W.-s.S. conceived and designed the experiments; J.-l. Lu, Y.-q.G., S.-y.L., Y.-y.Z., and B.L. performed the experiments; J.-l. Lu, X.-n.W., S.-w.F., J.Z., P.J., J.-L. Liang, P.W., S.-j.Z., F.-l.L., and X.Y. analyzed the data; J.-L. Liang, and J.-t.L. wrote the first draft of the manuscript; all authors revised the manuscript.

## Competing interests

The authors declare no competing interests.
