## [Peer Review File · Nature Communications]

Hidden diversity and potential ecological function of phosphorus acquisition genes in widespread terrestrial bacteriophagesReviewer #1 (Remarks to the Author):

The authors investigated the diversity of viruses carrying P-acquisition Auxiliary Metabolic Genes (AMGs) across 333 metagenomic samples from soil systems in China. The primary findings can be summarized as follows: (i) Within the study, 145 mVCs (metagenomic virus contigs) were identified, containing a total of 221 P-acquisition AMGs spanning 25 distinct functional genes; (ii) Notably, 40% of these AMGs were identified as novel; (iii) The functional analysis provided validation for the functional activity of the ppa gene; and (iv) The ubiquity of these P-acquisition AMGs is demonstrated on a global scale.

My concerns with the study are outlined below:

- 1) They must detail their methodology, some analysis such as ecological analysis (richness), and importantly validation of AMG identification (add genomic context validation).
- 2) Some figure captions are poorly written. A more detailed caption would help the reader understand the information. Moreover, to ensure the reader can understand the figures, make sure the figure quality and increase the font size.
- 3) Although there is no "gold standard" in doing auxiliary metabolic gene (AMG) analysis in viruses. However, I think they need to provide the reader with the validation data/annotation table of the viruses that carry P-acquisition AMGs, as this is the center point of this paper. Moreover, the author should justify why they use mVCs instead of vOTUs when identifying these AMGs.
- 4) While they did use contigs of ≥ 10 kb for vCONTACT v2 analysis, they should justify their choice in the case of using mVCs as opposed to vOTUs.
- 5) After examining the genome context of the candidate AMGs, particularly in Figure 4c, I have developed a concern. The AMGs are situated in a region devoid of any virus genes in proximity (virus hallmarks seem to be on the half part of the virus contig). It appears to be a cluster of P-acquisition genes, surrounded by prokaryotic and unknown genes. As I mentioned in the comment above, I suggest the authors present the annotations of all the viruses harboring these AMGs would offer readers invaluable insight. Further, checking the genomic context in AMG studies holds significant importance.
- 6) The authors have not kept up to date with the current development of the viromics pipelines, for example, DRAM-v. I would be interested to see the AMGs identified using more expert-curated pipelines. Moreover, I also suggest the author to also use HMM approach to identify the AMG, as this could potentially enhance the number, and accuracy of your identification analysis.

Below, I offer specific comments and suggestions:

Results:

Line 130: The author should provide a more detailed justification in line 130 for using metagenomic viral contigs (mVCs) instead of virus operational taxonomic units (vOTUs) for Auxiliary metabolic genes (AMGs) analysis. This would help readers better understand the author's choice and clarify any potential confusion.

Line 132: Include the annotation of the 145 mVCs in a supplementary table, this would make it easier for readers to access this important information.

Line 135: Of 221 P-acquisition AMGs, how many were unique gene families? Further, the author should clarify whether duplicates were observed among the 221 P-acquisition AMGs and provide more information on how this was assessed. This would help to ensure that readers have a clear understanding of the data and can interpret the results accurately.

Line 142: The authors should explain more about the term "ambiguous" in "ambiguous Caudovirales".

Lines 149-161: What is the significance of these findings?

Lines 151-154: 22 mVCs were shared among 4 different soil/habitat types and 100 mVCs were shared across all five types. What ecological significance does this sharing signify? How about those that are unique to specific types?

Line 186: For Figure 4a-d, do these viruses have circular genomes? If yes, can you direct me to this circular assessment? If not, please do not make it circular as it would be misleading the readers.

Line 213: You probably chose ppa gene for functional analysis because it showed conserved residues and 100% confidence values by Phyre2. However, other genes like ugpQ, pstS, and phoR also showed these characteristics. It might be a good idea to explain more about why you ended up choosing ppa gene instead of others.

Lines 222-255: It would be very informative to mention how many vOTU you have after dereplication (95% ID over 85% cov)? Also, please put the relative abundance table in the supplementary. And it is not clear to me how you calculate a diversity? Are you refereeing a diversity as Shannon's H?

Lines 232-242: I might have missed this, does the number of P-acquisition AMGs/phages across the habitat correlate to the P content in those habitats? In other words, the sentence explores whether the abundance of certain genes or viruses related to phosphorus acquisition is linked to the amount of phosphorus in the environment. Can you also provide the same data for the global distribution (lines 243-255)?

Line 235: The use of richness here as it referred to Figure 6a could be misleading. Should you just use "the number of"? furthermore, in this sentence, you also referred richness to in Figure 6b-f, which is the proportion (depicted in the relative abundance) of P AMGs across the sample. Please make sure to use the terms appropriately, while richness and relative abundance are related concepts in ecology, they refer to different aspects of a biological community's composition.

Line 257: Suggestion, it will be very useful to provide a schematic model of how the AMGs/viruses carrying the genes impact the soil ecosystems (studied soil system, i.e., farmland, forest, grassland, desert, and mine wasteland) and or host bacteria.

Line 258-259: For those (readers) unfamiliar with this analysis, I suggest you explain more about aggregated boosted tree (ABT) models is? And why used this model specifically?

In addition to functional and conserved domain analysis, I would suggest that the author conduct additional analysis, specifically calculating the ratio of non-synonymous to synonymous nucleotide differences (dN/dS) for the viral AMG families. This analysis is crucial as it provides insights into the selective pressures affecting these viruses. A dN/dS value of less than one would indicate that the virus is under selective pressure to retain a functional AMG, indicating its significance in maintaining viral fitness and potentially revealing its ecological role within its respective habitat.

Discussion:

Lines 347-376: Your explanation of how P-acquisition AMGs/phages, which carry these AMGs, operate in oceanic versus terrestrial systems, is not clear in terms of whether they function differently or similarly.

Lines 367-369: For viruses' impact on nitrogen (N) cycling via AMGs, you can also add this reference: <https://www.nature.com/articles/s41396-020-00825-6>

Line 373: While you have conducted host prediction analysis, there appears to be a deficiency in the explanation or meaningful exploration of the significant importance of the phage-host links within your system. Providing a comprehensive understanding of the relevance and implications of these phage-host connections would enhance the overall clarity and depth of your analysis.

Materials and Methods:

Line 426: Have you taken into consideration the possibility of utilizing the Hidden Markov Model (HMM) profile of P-acquisition genes for the analysis of identifying AMGs? Incorporating this approach could potentially enhance the number, and accuracy of your identification analysis.

Line 448: I have mentioned it before in my previous comments. I recommend the authors to put the annotation of the mVCs in the supplementary data.

Line 473: I do recommend doing taxonomy analysis only for vOTUs not for mVCs. This is due to vOTUs could potentially provide a more accurate representation of distinct viral species or populations. If you do not agree, please justify your reason.

Figures:

Although the guideline allowed the articles may have up to 10 display items (figures and/or tables). I think 9 figures in the main figures are too much. Please think through, which should be kept in the main and which can be moved to the supplementary.

I strongly believe that the figures presented in the main manuscript represent the most crucial results, while secondary figures can be included in the supplementary materials. These figures should be self-explanatory and accompanied by clear figure captions.

Figure 1a: Is the darker green color on the map indicative of areas where samples were taken, while the lighter green color represents areas where no samples were collected? This is very trivial, but you can indicate this in the figure caption to make it clearer.

Figure 1b: Could you please enlarge the inserted plot? Additionally, could you explain the significance of the teal color used in the plot?

Figure 1c: I would add % values to both the pie chart and the stacked bar plot.

Figure 1d: Do you consider this figure to be important? If yes, would it be possible to enlarge it? The current version has relatively small fonts and details.

Figure 2: Is this figure really important? I think this can go to supplementary.

Figure 3: Could you provide a brief explanation of what the term 'richness' in Fig. 3c refers to? This clarification will help alleviate any potential confusion.

Figure 4: I have previously commented on this figure above, but I'd like to reiterate my point. Do these viruses possess circular genomes? If they do, could you kindly direct me to the assessment of their circular nature? If not, I would suggest avoiding the portrayal of circular genomes, as this could potentially lead to confusion among readers.

Please explain the abbreviation of VOG in the caption as Virus Orthologous Groups

Figure 5: Explain the error bars (5b)

Figure 6: Again, explain briefly what "richness" here means.

Figure 7: The inserted panel is too small to read.

Supplementary:

Figure S6: Should the gene names be in italics? I would explain what the colors represent and the abbreviation of each gene.

Table S2: When possible, can you add information about the reactions each AMG facilitated?

Reviewer #2 (Remarks to the Author):

Liang et al. explored the diversity of P-acquisition genes encoded by phages on different terrestrial habitats. To that end, they analysed an impressive dataset of 333 soil metagenomes to reconstruct phage genomes. The authors also analysed public datasets and performed enzyme activity assays to complement their results. This is a timely and under explored topic, and this study has the potential to expand current knowledge on the field. However, the manuscript needs substantial improvement. Below I send my comments.

Major comments:

It is important to clarify to the broad audience what are phages. However, if you choose this term you need to be sure that you are addressing only bacterial viruses.

The Results section is really hard to follow. It needs a more focused and better structured writing (e.g. see specific comments below).

The analyses of public datasets should be better integrated with the whole story.

Why only two genes predicted to be involved in ppa had their activity tested?

It is important to show the readers more evidence that VIR_75 and VIR_90, which are the sequences harbouring AMG used in the enzymatic tests, are both bona fide viral sequence - i.e. presence of viral hallmark genes must be more evident. Also, they seem to be part of a same cluster as in Fig 4 VIR_75 is not shown?

Specific comments:

L109 - "comment"?

L130 - report the number of clusters; i.e. vOTUs.

L155-161 - this is not connected with the above text. Needs to show the reader how and why analysing public viral genomes support your results.

L163 - the wording is not adequate, better simplify to 'P-acquisition genes encoded by phages'? Furthermore, the way it is is not clear if the public genomes were/should be included in this analyses.

In general, utilisation of the term 'AMGs' is not helping much the reading flow as it usually refers to a broad spectrum of AMGs and the study focused on P-acquisition AMGs only. Adjusting accordingly would improve reading in many parts.

L184 - this section title could be more specific to better reflect its content.

L222 - this section is confusing as it overlaps with the first one.

L192 - reads better if starting by explaining the reader what are the 'four genes'.

L203 - remind the reader of what the other 21 genes refer to.

L222 - explain the reader what metrics is used for alfa diversity. Overall, this section doesn't read

nicely. Instead of writing ranges of alfa diversity why not showing a plot with statistical analyses comparing habitats?

L262 - clarify what is 'TP'.

L267 - how is this analysis related with the above (ABT)?

L273 - 'homologies of prokaryotes' is not appropriate wording. Could be 'homologous genes encoded by prokaryotes'. However, what/where is the evidence supporting this statement?

L285 - Host analysis results need to be better reported. Provide more details in terms of the number of linkages and what kind of hosts they are; e.g. are they involved in P-acquisition functions? This analysis should also be integrated with the phylogenetic analyses reported; e.g. L211.

L445 - isn't it the same trimming that is already performed by default by VS2?

L446 - also list the criteria.

L547 - CD-HIT at nucleotide level?

L473 - geNomad could also be used for straightforward taxonomic classification.

L588 - host predictions could be done using also other lines of evidence to improve the number of linkages identified; e.g. <https://doi.org/10.1016/j.chom.2020.08.003> or <https://doi.org/10.1371/journal.pbio.3002083>

Figure 4 'GC' is missing from b, c, and d

Reviewer #3 (Remarks to the Author):

This manuscript investigated P acquisition genes in a variety of environments across China, compared those genes found to published datasets, and experimentally validated the enzymatic activities of two P acquisition-related proteins encoded by AMGs. This work shows that 65% of the P-acquisition genes recovered in their metagenomes were detectable in published soil metagenome datasets, underscoring the ubiquity of these AMGs globally. Overall, the manuscript is intended to address an important research question that could be of interest to the general readers of Nature Communications. We listed some major and minor comments as follows.

Major comments:

1. Data interpretation:

- a. Line 136-137, I would be more careful about the interpretation of lytic/lysogenic annotations from VIBRANT. These classifications are based on the presence of integrases or whether the viral sequences were identified as integrated into a bacterial contig (<https://github.com/AnantharamanLab/VIBRANT/issues/16>). All the viral contigs that are not classified as lysogenic lifestyle will be assigned as lytic. That could be misleading especially for the contig fragments assembled from soil metagenomes.
- b. Percent confidence of the Phyre2 results is about model confidence. The percent of coverage and identity are the main parameters suggesting how similar the query structures are to the reference structures in the database. The way that the results were introduced in lines 201-202 could be a bit misleading. Another question...Could the active sites be located on the modeled structures?
- c. It is surprising to see a rarefaction curve of the viral sequences detected from soil metagenomes (even with the sampling effort in this study) has reached a plateau. Are the mVCs de-replicated? A

total of 35552 viral contigs from 333 soil metagenomes result in an average of 106 viral contigs per sample. This may suggest a low recovery rate in viral sequences and partially explain the rarefaction curve. It could be good to acknowledge this limitation.

d. Line 395, potentially overstating the role of P acquisition genes in virus-host co-evolution.

e. Recommend highlighting the experimentally validated enzymatic activities of two P acquisition-related proteins more to emphasize the importance of the finding. It would be helpful for readers for you to separate that into its own section, dividing results validated computationally from results validated experimentally, as it gets lost in that section.

2. Methodology:

a. it is unclear how the taxonomy of mVCs was assigned. Based on the method section, it was assigned by blasting mVC proteins against the NCBI viral proteins with known taxonomy. So the assignment is based on a single protein hit even given not all proteins are phylogenetic markers? The protein sharing network (fig1d) also showed that viral contigs with different 'assigned' taxa were grouped into the same cluster. it may indicate some disagreement of taxonomic assignment using the protein hit method VS the protein sharing method.

b. AMGs are first annotated by general databases (e.g., KEGG) as they share sequence similarities to the non-viral metabolic genes. Then mapping the short metaG or metaT reads (though high-quality)... how confident these AMG-mapped reads are not from the non-viral metabolic genes? Have the authors cross-checked the mapping results against the non-viral genes? If it is not carefully handled, it will affect some of the main discussions using quantitative results.

Minor line edits

3. Line 106, P is not the only growth limiting factor for bacteria.

4. Lines 42 and 273, homologs instead of homologies.

5. Grammar corrections in regards to the use of commas (see the Oxford Comma) necessary throughout the Results and Discussion section (Lines 148, 151, 197, 214, 226, 240, 246, 264, 332, etc)

6. Line 109, use of the word "comment" not understandable, perhaps you meant "similar"?

7. Line 112, it may be not accurate to say, 'only one study'. There are other soil studies that have reported the detection of P-related AMGs including phoH. It is also risky to claim a study is the only one as it takes some time for a manuscript to publish and we are not sure if there will be new papers out. Please consider paraphrasing the sentence to be more accurate and avoid the uncertainty.

8. Line 142, 'ambiguous Caudovirales' is not a common way to describe a taxon. Please define it in front.

9. Line 144-145, please note that Siphoviridae, Myoviridae and Podoviridae have been abolished by the recent release of ICTV.

10. Fig2 plots data of total mVC or P-acquisition phages? Please make the figure legend consistent with the text in lines 149-154.

11. Please correct the spelling of 'i.e.' throughout the manuscript. add ', 'after 'i.e.'.

12. Line 172, grammar (a dramatic increase)

13. Line 177, it is unclear what is the 'highest'. Perhaps the most prevalent? The highest occurrence across soil samples?

14. Line 186 grammar (arrangements)

15. Figure 4: The branches colored in black represent the phylogenetic placement of the reference bacterial sequences? Please make it clear in the legend key or text. Some of the reference clades are too big to clearly visualize the distribution of the AMGs. please consider either reducing the number of reference sequences or collapsing some of these clades.

16. Line 208, syntax and grammar (completely lacking)

17. Line 216-217, are the two PPase families corresponding to the certain clades of the tree in fig. 4e?

18. Line 225-227, were the numbers of the P-acquisition phages counted after removing the AMG proteins lacking the known active sites?

19. Line 227 spelling (and)

20. Line 247-250, the same P-acquisition mVCs and the P-related AMGs were also detected in the global soil metagenomes. I am wondering if these mVCs in the global dataset also carried the same P-

related AMGs as they were detected in the dataset curated in this study. It may be interesting to test if soil viruses harbor the same AMGs regardless of the different sites/terrestrial systems.

21. Line 269-271, separating the % from the land type made this hard to read. Suggest reordering (i.e. mine wasteland (19.4%), farmland (19.2%), etc.).

22. Line 287, please make it clear what the 'transcript ratio of phage:host pairs' is. Transcript coverage across the whole genomes/contigs or the P-related gene homologs?

23. Line 304-305, as mentioned, it is not accurate (e.g., DOI: <https://doi.org/10.1128/msystems.00076-18>, <https://doi.org/10.1038/s41396-022-01188-w>, <https://doi.org/10.1038/s41396-022-01188-w> and more examples in sediments...)

24. Lines 327 & 343, ubiquity

25. Line 322, as PPases are mainly involved in DNA, RNA, and protein syntheses. How confident are they classified as AMGs? If the confident assignment is based on the position of the ppa genes away from the other nucleotide metabolism genes, have the authors observed a clear differentiation of the AMG ppa and non-AMG ppa?

26. Discussion section, be consistent with italicizing genes or not (lines 329-337 in particular)

27. 355 grammar, components

28. Line 370, they rather than it

29. Lines 383-385, tenses of verbs, grammar

30. Line 390, arising.. or acquired

31. Lines 392-394. Reduced intracellular content of P? unclear what you are referring to in this sentence.

32. Line 397, exerts

33. Line 401, increasingly.

34. Line 404, scales

35. Line 440-441, grammar check.

36. Line 459, please write out the three criteria.

37. Line 463, checkV database? IMG/VR is one of the public databases mentioned in line 156. Please double-check.

38. Please provide version info of each tool used (e.g., MAFFT and IQ-TREE).

Reviewer #4 (Remarks to the Author):

Responses to the Reviewers' comments

Responses to Reviewer #1's comments

Reviewer #1 (Remarks to the Author):

The authors investigated the diversity of viruses carrying P-acquisition Auxiliary Metabolic Genes (AMGs) across 333 metagenomic samples from soil systems in China. The primary findings can be summarized as follows: (i) Within the study, 145 mVCs (metagenomic virus contigs) were identified, containing a total of 221 P-acquisition AMGs spanning 25 distinct functional genes; (ii) Notably, 40% of these AMGs were identified as novel; (iii) The functional analysis provided validation for the functional activity of the *ppa* gene; and (iv) The ubiquity of these P-acquisition AMGs is demonstrated on a global scale.

Response: We thank this reviewer for acknowledging the merits of our manuscript.

My concerns with the study are outlined below:

1) They must detail their methodology, some analysis such as ecological analysis (richness), and importantly validation of AMG identification (add genomic context validation).

Response: Thank you for your constructive comments. In order to provide more details on our methodology, revisions have been made on the Methods section of the original manuscript (OM). The major revisions included (but were not limited to): (1) the subsection regarding AMG identification in the OM has been almost completely rewritten to explain in detail how we validated the AMG identification, with two new supplementary tables (i.e., Supplementary Tables 5 and 6) being added to the revised manuscript (RM) to provide the genomic contexts of the identified P-acquisition AMGs annotated by VIBRANT and DRAM-v (RM: Lines 526–564); (2) we have changed “richness” into “the number of ...” to avoid confusion and explained how “the number

of P-acquisition vOTUs (or AMGs) detected in a given sample” was evaluated (RM: Lines 267, 279, and 954); and (3) a new subsection has been added to explain the ABT analysis (an important ecological analysis) we made (RM: Lines 701–715). As a result, the length of the Methods section has been extended from 2015 words in the OM to 2751 words in the RM. We really hope that these revisions have considerably improved the quality of the Methods section.

2) Some figure captions are poorly written. A more detailed caption would help the reader understand the information. Moreover, to ensure the reader can understand the figures, make sure the figure quality and increase the font size.

Response: Agreed. In the RM, we have rewritten most of the figure captions to provide a more detailed caption for each figure. Meanwhile, we have tried to improve the figure quality and increase the font size to ensure that the readers can understand the figures. Please refer to Figures 1–8 of the RM for details on our revisions.

3) Although there is no “gold standard” in doing auxiliary metabolic gene (AMG) analysis in viruses. However, I think they need to provide the reader with the validation data/annotation table of the viruses that carry P-acquisition AMGs, as this is the center point of this paper. Moreover, the author should justify why they use mVCs instead of vOTUs when identifying these AMGs.

Response: Agreed. According to this comment and your specific advice listed below, we have used DRAM-v in DRAM (v1.2.0) to validate the potential P-acquisition AMGs located in the vOTUs recovered from our metagenomes, with the genomic contexts of the identified P-acquisition AMGs annotated by VIBRANT and DRAM-v being provided in two new supplementary tables (i.e., Supplementary Tables 5 and 6 in the RM). The main results of the validation were summarized as follows:

(1) According to DRAM-v, 89 genes with auxiliary scores of 1–3 and AMG flag of –M and/or –F were considered as AMGs. (2) Eight genes with auxiliary scores of 1–3 and AMG flag of –T were also considered as AMGs, because the vOTUs harboring

these genes were validated as phages by the three methods used in this study. (3) As per DRAM-v, a gene is not considered as an AMG if it is in a row of three metabolic genes (with AMG flag of -B and an auxiliary score of 4). However, the genome of one isolated bacteriophage, *Campylobacter* phage A18a (GCA_002956935.1), harbored *phoU* and *pstBACS* in a row (Figure A1 presented in the next page of this document), suggesting that *pst* gene cluster can actually exist in a phage genome. Therefore, the *pstSCAB* gene clusters (with auxiliary scores of 4) present in two vOTUs (i.e., vOTU47 and vOTU60; Figure A1) were considered as AMGs. (4) Each of the 17 AMG kinds identified in our metagenomes was represented by at least one phage gene sequence with an auxiliary score of 1 or 2. For more details, please refer to RM: Lines 182–193 and 549–564.

As to the analysis of mVCs in the OM, we did so because we were inspired by a previous paper regarding inorganic sulfur auxiliary metabolism in phages (Kieft *et al.*, 2021, *Nature Communications*, 12, 3503). Note that, in the OM, we only used mVCs in AMG identification, with vOTUs (dereplicated at 95% identity and 85% alignment fraction) having been used in those subsequent analyses (e.g., metagenomic mapping, metatranscriptomic mapping, and host prediction). Nonetheless, for a more accurate representation of distinct viral populations, we have used vOTUs instead of mVCs in all analyses of the RM (including AMG identification, taxonomy analysis, ecological analysis, etc.).

Figure A1. Genome organization diagrams of two P-acquisition vOTUs recovered from our soil metagenomes and one public isolated phage genome. They all encode *pstSCAB*. Predicted open reading frames are colored according to VIBRANT and DRAM-v annotation functions. vOTU, viral operational taxonomic unit. VOG, virus orthologous groups.

4) While they did use contigs of ≥ 10 kb for vConTACT v2 analysis, they should justify their choice in the case of using mVCs as opposed to vOTUs.

Response: As mentioned in our response to your comment above, for a more accurate representation of distinct viral populations, we have used vOTUs instead of mVCs in all analyses of the RM.

5) After examining the genome context of the candidate AMGs, particularly in Figure 4c, I have developed a concern. The AMGs are situated in a region devoid of any virus genes in proximity (virus hallmarks seem to be on the half part of the virus contig). It appears to be a cluster of P-acquisition genes, surrounded by prokaryotic and unknown genes. As I mentioned in the comment above, I suggest the authors present the annotations of all the viruses harboring these AMGs would offer readers invaluable insight. Further, checking the genomic context in AMG studies holds significant importance.

Response: As mentioned above, we have used DRAM-v in DRAM (v1.2.0) to validate the potential P-acquisition AMGs located in the vOTUs recovered from our metagenomes. The genomic contexts of the identified P-acquisition AMGs annotated by VIBRANT and DRAM-v have been provided in Supplementary Tables 5 and 6 in the RM. Notably, the identified P-acquisition AMGs in the RM (e.g., those presented in new Figure 2c) were flanked by viral hallmark or viral-like genes on both sides. In contrast, the candidate AMGs shown in the Figure 4c of the OM were no longer considered as AMGs according to the DRAM-v results and have been thus excluded from the RM.

6) The authors have not kept up to date with the current development of the viromics pipelines, for example, DRAM-v. I would be interested to see the AMGs identified using more expert-curated pipelines. Moreover, I also suggest the author to also use HMM approach to identify the AMG, as this could potentially enhance the number, and accuracy of your identification analysis.

Response: Thanks for your constructive suggestions. In the RM, DRAM-v and HMM approach have been used. Given that the main results from the DRAM-v analysis have been summarized above, we summarized the main results from the HMM search against KEGG database (along with those from the Diamond BLASTP against KEGG database) as follows:

Using HMM search and Diamond BLASTP against KEGG database, we identified 188 and 216 potential P-acquisition AMGs from the vOTUs recovered from our metagenomes, respectively. After eliminating duplicates, 257 potential P-acquisition AMGs were obtained. Among the 168 vOTUs encoding these 257 potential P-acquisition AMGs, 115 were further validated as phages by three methods: (i) VIBRANT (v1.2.1; default parameters); (ii) SOP used in Sullivan Lab for viral identification based on VirSorter2 (v2.2.3) and CheckV (v0.9.0; dx.doi.org/10.17504/protocols.io.bwm5pc86); and (iii) manual curation based on VIBRANT annotations (i.e., KEGG, Pfam, and VOG databases) with three established criteria (please see Methods in the RM for details). To validate the potential P-acquisition AMGs of these 115 vOTUs, we used DRAM-v and identified 105 P-acquisition AMGs, which were encoded by 75 vOTUs (Supplementary Tables 5 and 6 in the RM). For more details, please refer to RM: Lines 119–126 and 527–564.

Below, I offer specific comments and suggestions:

Results:

Line 130: The author should provide a more detailed justification in line 130 for using metagenomic viral contigs (mVCs) instead of virus operational taxonomic units (vOTUs) for Auxiliary metabolic genes (AMGs) analysis. This would help readers better understand the author's choice and clarify any potential confusion.

Response: As mentioned above, we have used vOTUs instead of mVCs for AMG analysis (and other relevant analyses) in the RM. Overall, we recovered 37,896 mVCs from our metagenomic datasets and obtained 35,552 vOTUs after dereplication (at 95% identity and 85% alignment fraction). Thus, the related sentence has been changed into

“A total of 35,552 viral operational taxonomic units (vOTUs) were recovered from our 333 soil metagenomes (Figure 1 and Supplementary Table 1).” (RM: Lines 118 and 119).

Line 132: Include the annotation of the 145 mVCs in a supplementary table, this would make it easier for readers to access this important information.

Response: In the RM, using the updated viromics pipelines, we identified a total of 75 vOTUs that encoded P-acquisition genes. The annotations of these P-acquisition vOTUs by VIBRANT and DRAM-v have been provided in Supplementary Tables 5 and 6 of the RM, respectively.

Line 135: Of 221 P-acquisition AMGs, how many were unique gene families? Further, the author should clarify whether duplicates were observed among the 221 P-acquisition AMGs and provide more information on how this was assessed. This would help to ensure that readers have a clear understanding of the data and can interpret the results accurately.

Response: In the RM, using the updated viromics pipelines, we identified a total of 75 vOTUs that encoded 105 P-acquisition genes. These P-acquisition AMGs were non-redundant (dereplicated at 95% sequence similarity and 90% alignment coverage). They spanned 17 distinct functional genes (i.e., 17 gene kinds) and belonged to 18 unique gene families, as one of the 17 gene kinds (i.e., *ppa*) included genes from two nonhomologous families (i.e., soluble Family I and Family II). For more details, please refer to RM: Lines 140–148, 198–200, 211–215, and 241–244).

Line 142: The authors should explain more about the term “ambiguous” in “ambiguous Caudovirales”.

Response: The term “ambiguous *Caudovirales*” has been deleted in the RM. Because, we have used geNomad (v1.7.0; according to the Reviewer #2’ suggestion) and PhaGCN (v2.0) for viral taxonomic classification in the RM, adopting the updated

nomenclature released by the International Committee on Taxonomy of Viruses (ICTV). The new results on viral taxonomy profiles are presented in RM: Lines 130–139.

Lines 149-161: What is the significance of these findings?

Response: According to the new results obtained in the RM, the two paragraphs have been almost completely rewritten and assigned into two different subsections of the Results section (RM: Lines 155–169 and 261–264). In the Discussion section, we have tried to point out the significance of the relevant findings as follows: (1) the widespread of certain P-acquisition phages and the related AMGs indicates that they are involved in the major metabolic strategies by which terrestrial phages could assist their hosts to cope with P-limited conditions (RM: Lines 409–411 and 459–462); and (2) the identification of seven P-acquisition AMG kinds not reported in previous studies suggests that public databases should not be overlooked in future studies on viral auxiliary metabolism (RM: Lines 404–408).

Lines 151-154: 22 mVCs were shared among 4 different soil/habitat types and 100 mVCs were shared across all five types. What ecological significance does this sharing signify? How about those that are unique to specific types?

Response: As mentioned above, the widespread of certain P-acquisition phages and the related AMGs indicates that they are involved in the major metabolic strategies by which terrestrial phages could assist their hosts to cope with P-limited conditions. With regard to those that are unique to specific types, we believe that they likely reflect the habitat preferences of certain phages. In the RM, we preferred not to discuss this point, given that it seems beyond the focus of this study a little bit.

Line 186: For Figure 4a-d, do these viruses have circular genomes? If yes, can you direct me to this circular assessment? If not, please do not make it circular as it would be misleading the readers.

Response: Sorry for the confusion. No, these viruses did not have circular genomes.

and they were just visualized as circulars with the endpoints being indicated by black lines (as mentioned in the caption of Figure 4 in the OM). According to your suggestion, we have presented the gene arrangements of four representative vOTUs in a line (RM: Figure 2c).

Line 213: You probably chose *ppa* gene for functional analysis because it showed conserved residues and 100% confidence values by Phyre2. However, other genes like *ugpQ*, *pstS*, and *phoR* also showed these characteristics. It might be a good idea to explain more about why you ended up choosing *ppa* gene instead of others.

Response: Thanks for the suggestion. In the RM, we have added the contents about functional validation of another P-acquisition AMG (i.e., *phoD*). Meanwhile, we have explained why we chose *ppa* and *phoD* genes for functional analysis in the RM as follows: “To test whether the identified P-acquisition AMGs can encode functionally active proteins, representative phage genes that were involved in inorganic P solubilization (*ppa*) or organic P mineralization (*phoD*) were selected for functional validation via *in-vitro* assays. The functional validation of AMGs related to P transportation and P regulation (e.g., *pstS* and *phoR*, respectively) was not considered in this study for two reasons. First, the function of a phage *pstS* has been validated previously (Zhao et al, 2022). Second, the function of a regulatory gene (e.g., *phoR*) is very difficult to verify in an *in-vitro* system.”. For more details, please refer to RM: Lines 233–259.

Lines 222-255: It would be very informative to mention how many vOTU you have after dereplication (95% ID over 85% cov)? Also, please put the relative abundance table in the supplementary. And it is not clear to me how you calculate α diversity? Are you refereeing α diversity as Shannon’s H?

Response: In the RM, we recovered a total of 75 P-acquisition vOTUs (dereplicated at 95% identity and 85% alignment fraction) from our metagenomes. As suggested, we have provided the relative abundance table of these 75 vOTUs in the supplementary

(i.e., Supplementary Table 14 in the RM). In the OM, we had referred alpha diversity as the number of vOTUs (or AMGs). To avoid confusion, we have used “the number of vOTUs (or AMGs)” instead of “richness or alpha diversity” throughout the RM.

Lines 232-242: I might have missed this, does the number of P-acquisition AMGs/phages across the habitat correlate to the P content in those habitats? In other words, the sentence explores whether the abundance of certain genes or viruses related to phosphorus acquisition is linked to the amount of phosphorus in the environment. Can you also provide the same data for the global distribution (lines 243-255)?

Response: In the OM, we had reported that the normalized gene coverages of P-acquisition AMGs were positively correlated with soil available P contents (OM: Lines 267 - 269). In the RM, we have found that the numbers of P-acquisition vOTUs/AMGs were significantly ($P < 0.001$) correlated with soil total P/available P contents (RM: Lines 268 - 271 and 280 - 283).

In order to address your concern about those results based on global data, we have downloaded a public dataset of the global total soil P concentrations (He *et al.*, 2021, *Earth System Science Data*, 13, 5831–5846) and revealed a significant ($P = 0.005$) correlation between the number of P-acquisition vOTUs and soil total P contents (RM: Lines 300–304). By the way, we found that no public dataset of the global bioavailable soil P concentrations has been published so far.

Line 235: The use of richness here as it referred to Figure 6a could be misleading. Should you just use “the number of”? furthermore, in this sentence, you also referred richness to in Figure 6b-f, which is the proportion (depicted in the relative abundance) of P AMGs across the sample. Please make sure to use the terms appropriately, while richness and relative abundance are related concepts in ecology, they refer to different aspects of a biological community's composition.

Response: Thanks for the suggestion. To avoid confusion, we have changed the “richness” into “the number of” throughout the RM. In the OM, we had also referred

“richness” to in Figure 6b-f, as the data on the number of P-acquisition AMGs detected in individual study sites had been shown on the X-axis of the right-hand side of each panel. Meanwhile, in the RM, we have clarified that the proportions shown in Figure 4b-4f (i.e., those shown in Figure 6b-6f of the OM) are “the relative abundances of the four categories of P-acquisition AMGs” (RM: Lines 283 – 291 and 954–960).

Line 257: Suggestion, it will be very useful to provide a schematic model of how the AMGs/viruses carrying the genes impact the soil ecosystems (studied soil system, i.e., farmland, forest, grassland, desert, and mine wasteland) and or host bacteria.

Response: According to your suggestion, we have proposed a schematic model of how terrestrial P-acquisition phages can impact the P-acquisition processes of their hosts (i.e., Figure 8 in the RM). For details, please refer to RM: Lines 371 – 386.

Line 258-259: For those (readers) unfamiliar with this analysis, I suggest you explain more about aggregated boosted tree (ABT) models is? And why used this model specifically?

Response: Thanks for your suggestion. In the RM, a new subsection named “ABT analysis” has been added to the Methods section to explain briefly what is ABT analysis and why (and how) we used it in this study (RM: Lines 701 – 715).

In addition to functional and conserved domain analysis, I would suggest that the author conduct additional analysis, specifically calculating the ratio of non-synonymous to synonymous nucleotide differences (dN/dS) for the viral AMG families. This analysis is crucial as it provides insights into the selective pressures affecting these viruses. A dN/dS value of less than one would indicate that the virus is under selective pressure to retain a functional AMG, indicating its significance in maintaining viral fitness and potentially revealing its ecological role within its respective habitat.

Response: Thanks for the suggestion. In the RM, we have tried to calculate dN/dS for individual P-acquisition AMG kinds, and shown that the calculated dN/dS values were

less than one. For details, please refer to RM: Lines 223 – 231, 465 – 470, and 604 – 616.

Discussion:

Lines 347-376: Your explanation of how P-acquisition AMGs/phages, which carry these AMGs, operate in oceanic versus terrestrial systems, is not clear in terms of whether they function differently or similarly.

Response: In the RM, the relevant contents have been almost completely rewritten to discuss one of the three major metabolic strategies that may be employed by terrestrial phages to assist their hosts to cope with P-limited conditions (RM: Lines 434–453).

Lines 367-369: For viruses' impact on nitrogen (N) cycling via AMGs, you can also add this reference: <https://www.nature.com/articles/s41396-020-00825-6>

Response: Done as suggested (RM: Line 453).

Line 373: While you have conducted host prediction analysis, there appears to be a deficiency in the explanation or meaningful exploration of the significant importance of the phage-host links within your system. Providing a comprehensive understanding of the relevance and implications of these phage-host connections would enhance the overall clarity and depth of your analysis.

Response: Thanks for the suggestion. In the RM, to address phage-host connections, we have followed the Reviewer #2's suggestion and used iPHoP (v1.3.3) for host prediction. The iPHoP attempts to link viral sequences to a host taxon based on multiple criteria: (i) direct sequence comparison to host genomes and CRISPR spaces, (ii) overall nucleotide composition to host genomes, and (iii) comparison to viruses with known hosts. Overall, we found that 35% (26/75) of the P-acquisition vOTUs identified in our metagenomes could be associated with a total of 28 hosts, which spanned 10 bacterial and one archaeal phyla (RM: Figure 7a and Supplementary Table 17). For more details, please refer to RM: Lines 330–346.

The newly obtained results on phage-host links, along with the abovementioned dN/dS values and the widespread distribution of certain P-acquisition AMGs, have been employed to discuss the potential effects of chronic P scarcity on phage–host evolution (RM: Lines 465–486).

Materials and Methods:

Line 426: Have you taken into consideration the possibility of utilizing the Hidden Markov Model (HMM) profile of P-acquisition genes for the analysis of identifying AMGs? Incorporating this approach could potentially enhance the number, and accuracy of your identification analysis.

Response: As mentioned above, we have used HMM search (HMMER 3.3.2) against KOfam database to identify P-acquisition AMGs in the RM. For more details, please refer to our response to your related comment above.

Line 448: I have mentioned it before in my previous comments. I recommend the authors to put the annotation of the mVCs in the supplementary data.

Response: As mentioned above, we have provided the genomic contexts of the identified P-acquisition vOTUs annotated by VIBRANT and DRAM-v in the supplementary data (i.e., Supplementary Tables 5 and 6 in the RM).

Line 473: I do recommend doing taxonomy analysis only for vOTUs not for mVCs. This is due to vOTUs could potentially provide a more accurate representation of distinct viral species or populations. If you do not agree, please justify your reason.

Response: As mentioned above, in the RM, we have followed your advice and performed taxonomy analysis only for vOTUs.

Figures:

Although the guideline allowed the articles may have up to 10 display items (figures and/or tables). I think 9 figures in the main figures are too much. Please think through,

which should be kept in the main and which can be moved to the supplementary.

Response: According to the comment, Figure 2 in the OM has been moved to the supplementary, while Figures 3 and 4 in the OM have been integrated into a new figure (i.e., Figure 2 in the RM). After these adjustments, the RM now has eight main figures: seven figures are used to illustrate the most crucial results we want to show to the readers, and one figure is used to propose a schematic model (as suggested by this reviewer). We hope you will agree that the number of figures in the RM is OK.

I strongly believe that the figures presented in the main manuscript represent the most crucial results, while secondary figures can be included in the supplementary materials. These figures should be self-explanatory and accompanied by clear figure captions.

Response: Agreed. In the RM, most of the figure captions have been almost completely rewritten to make sure that all figures are self-explanatory and accompanied by clear figure captions.

Figure 1a: Is the darker green color on the map indicative of areas where samples were taken, while the lighter green color represents areas where no samples were collected? This is very trivial, but you can indicate this in the figure caption to make it clearer.

Response: Yes, you are right that the darker green color on the map is indicative of areas where samples were taken and the lighter green color represents areas where no samples were collected. In the RM, we have indicated this as suggested in the figure caption to make it clearer (RM: Lines 922 and 923).

Figure 1b: Could you please enlarge the inserted plot? Additionally, could you explain the significance of the teal color used in the plot?

Response: Thanks for the suggestions. In the RM, the inserted plot has been enlarged (please see the inserted plot in Figure 1b). Meanwhile, we have indicated in the figure caption that the teal color used in the Figure 1b represents 500 iterations (sample order randomizations) to facilitate the readers to understand its meaning (RM: Lines 925 and

926).

Figure 1c: I would add % values to both the pie chart and the stacked bar plot.

Response: Done as suggested (RM: Figure 1c).

Figure 1d: Do you consider this figure to be important? If yes, would it be possible to enlarge it? The current version has relatively small fonts and details.

Response: In the RM, we have deleted Figure 1d, because new methods were employed to analyze the viral taxonomy profiles.

Figure 2: Is this figure really important? I think this can go to supplementary.

Response: According to your suggestion, this figure has been moved to supplementary in the RM (Figure S22).

Figure 3: Could you provide a brief explanation of what the term 'richness' in Fig. 3c refers to? This clarification will help alleviate any potential confusion.

Response: In order to avoid confusion, the term “richness” in Fig. 3c in the OM has been changed into “Gene kind” (please see Figure 2b in the RM).

Figure 4: I have previously commented on this figure above, but I'd like to reiterate my point. Do these viruses possess circular genomes? If they do, could you kindly direct me to the assessment of their circular nature? If not, I would suggest avoiding the portrayal of circular genomes, as this could potentially lead to confusion among readers. Please explain the abbreviation of VOG in the caption as Virus Orthologous Groups

Response: As mentioned in our response to your related comment above, these viruses did not have circular genomes, and they were just visualized as circular with the endpoints being indicated by black lines (as indicated in the figure caption of Figure 4 in the OM). To avoid confusion, we have presented the gene arrangements of four representative vOTUs in a line (Figure 2c in the RM) and explained the abbreviation of

VOG in the figure caption as Virus Orthologous Groups (RM: Lines 938 and 939).

Figure 5: Explain the error bars (5b)

Response: Done as suggested (RM: Line 950).

Figure 6: Again, explain briefly what “richness” here means.

Response: The term “richness” has been changed into “the number of” (RM: Line 954).

Figure 7: The inserted panel is too small to read.

Response: Agreed. Based on the updated results, the inserted panel has been changed into an independent panel (RM: Figure 5c).

Supplementary:

Figure S6: Should the gene names be in italics? I would explain what the colors represent and the abbreviation of each gene.

Response: The gene names are shown in italics. Meanwhile, we have explained the colors represent and the abbreviation of each gene (RM: the caption of Supplementary Figure 19).

Table S2: When possible, can you add information about the reactions each AMG facilitated?

Response: Done as suggested (RM: Supplementary Table 2).

Responses to Reviewer #2’s comments

Reviewer #2 (Remarks to the Author):

Liang et al. explored the diversity of P-acquisition genes encoded by phages on different

terrestrial habitats. To that end, they analysed an impressive dataset of 333 soil metagenomes to reconstruct phage genomes. The authors also analysed public datasets and performed enzyme activity assays to complement their results. This is a timely and under explored topic, and this study has the potential to expand current knowledge on the field. However, the manuscript needs substantial improvement. Below I send my comments.

Response: We thank this reviewer for acknowledging the merits of our manuscript. We have tried our best to address each of your comments. Below, please find our point-by-point responses to your comments. We really hope you will find that our manuscript is improved considerably.

Major comments:

It is important to clarify to the broad audience what are phages. However, if you choose this term you need to be sure that you are addressing only bacterial viruses.

Response: Thanks for the reminder. In the revised manuscript (RM), we have clarified what are phages (RM: Line 80). We have preferred to use this term, because we found that only one out of the 75 P-acquisition vOTUs identified in our metagenomes was predicted to have hosts from the Archaea domain. Note that this vOTU was also associated with hosts from the Bacteria domain (RM: Lines 337–339).

The Results section is really hard to follow. It needs a more focused and better structured writing (e.g. see specific comments below).

Response: According to this comment and your specific comments below, we have made several adjustments on the structure of the Results section to achieve a more focused and better structured writing. Below, please find our point-by-point responses to your comments.

The analyses of public datasets should be better integrated with the whole story.

Response: Agreed. In order to better integrate the analyses of public datasets with the whole story, the following three major revisions have been made: (1) we pointed out clearly the main aim of such analyses (RM: Lines 156–158); (2) the main results of such analyses were introduced in a new independent subsection named “P-acquisition phages and AMGs recovered from public viral databases” (RM: Lines 155–169), with three representative P-acquisition phages recovered from public viral databases being shown in Figure 2c (for details, please refer to RM: Lines 178–181 and 186–188); and (3) the important implications of the results from such analyses for understanding P-acquisition phages and AMGs were discussed along with the results from those analyses based on our metagenomes (RM: Lines 399–408, 411–419, 421–424, and 443–453).

Why only two genes predicted to be involved in *ppa* had their activity tested?

Response: In the RM, we have added the contents about the functional validation of another P-acquisition AMG (i.e., *phoD*). Meanwhile, we have explained why we chose *ppa* and *phoD* genes for functional analysis in the RM as follows: “To test whether the identified P-acquisition AMGs can encode functionally active proteins, representative phage genes that were involved in inorganic P solubilization (*ppa*) or organic P mineralization (*phoD*) were selected for functional validation via *in-vitro* assays. The functional validation of AMGs related to P transportation and P regulation (e.g., *pstS* and *phoR*, respectively) was not considered in this study for two reasons. First, the function of a phage *pstS* has been validated previously (Zhao et al, 2022). Second, the function of a regulatory gene (e.g., *phoR*) is very difficult to verify in an *in-vitro* system.”. For more details, please refer to RM: Lines 233–259.

It is important to show the readers more evidence that VIR_75 and VIR_90, which are the sequences harbouring AMGs used in the enzymatic tests, are both bona fide viral sequence - i.e. presence of viral hallmark genes must be more evident. Also, they seem to be part of a same cluster as in Fig 4 VIR_75 is not shown?

Response: Thanks for the comment. In the RM, the two *ppa* genes selected for the

enzymatic tests were encoded by vOTU1 and vOTU54, respectively (RM: Lines 241–244). The gene arrangement of vOTU54 has been shown in Figure 2c, while that of vOTU1 has been shown in Supplementary Figure 21d. The two *ppa* genes were flanked by viral hallmark or viral-like genes on both sides (RM: Figure 2c and Supplementary Figure 21d), supporting their affiliations to viruses.

Specific comments:

L109 - “comment”?

Response: Changed into “similar” (RM: Line 96).

L130 - report the number of clusters; i.e. vOTUs.

Response: Done as suggested (RM: Line 118).

L155-161 - this is not connected with the above text. Needs to show the reader how and why analysing public viral genomes support your results.

Response: Agreed. In order to better integrate the analyses of public datasets with the whole story, three major revisions have been made in the RM. For more details, please refer to our response to your related comment above.

L163 - the wording is not adequate, better simplify to ‘P-acquisition genes encoded by phages’? Furthermore, the way it is is not clear if the public genomes were/should be included in this analyses.

Response: Thanks for the comment. In the RM, the results on P-acquisition phages and AMGs identified in our metagenomes have been presented in a new subsection named “P-acquisition phages and AMGs identified in soil metagenomes from across China” (RM: Lines 117–153), while those results of the analyses of public viral datasets have been shown in another new subsection named “P-acquisition phages and AMGs recovered from public viral databases” (RM: Lines 155–169). In this way, the results

based on our metagenomes and those based on public viral datasets are presented separately, which should be helpful to avoid confusion.

In general, utilisation of the term ‘AMGs’ is not helping much the reading flow as it usually refers to a broad spectrum of AMGs and the study focused on P-acquisition AMGs only. Adjusting accordingly would improve reading in many parts.

Response: Agreed. We have changed “AMGs” into “P-acquisition AMGs” throughout the RM.

L184 - this section title could be more specific to better reflect its content.

Response: In the RM, this subsection of the Results section has been named as “Conserved amino acid residues in proteins encoded by P-acquisition AMGs” (RM: 195–221). Meanwhile, in order to make sure that the subsection title is more specific to better reflect its content, another additional adjustment has been made: the results from phylogenetic analysis and the functional validation of P-acquisition AMGs have been excluded from this subsection.

L222 - this section is confusing as it overlaps with the first one.

Response: In the RM, the contents regarding geographic distribution of P-acquisition phages have been excluded from the first subsection of the Results section and integrated into a new subsection named “Geographic distribution and community composition of P-acquisition phages and AMGs” (RM: Lines 261–313).

L192 - reads better if starting by explaining the reader what are the ‘four genes’.

Response: Thanks for the suggestion. In the RM, the related sentences have been completely rewritten (RM: Lines 196–205).

L203 - remind the reader of what the other 21 genes refer to.

Response: Done as suggested (RM: Lines 206 and 207).

L222 - explain the reader what metrics is used for alfa diversity. Overall, this section doesn't read nicely. Instead of writing ranges of alfa diversity why not showing a plot with statistical analyses comparing habitats?

Response: Thanks for the comment. In our manuscript, the number of P-acquisition vOTUs (AMGs) detected in a given sample was used for the alpha diversity of P-acquisition vOTUs (AMGs) in that sample. In order to avoid confusion, we have replaced “alpha diversity” with “the number of” throughout the RM. Meanwhile, according to your suggestion, a new plot has been added to show the statistical analyses comparing habitat types (RM: Supplementary Figure 23a). For more details, please refer to RM: Lines 264–268.

L262 - clarify what is 'TP'.

Response: Done as suggested (RM: Line 269).

L267 - how is this analysis related with the above (ABT)?

Response: This analysis is not related with ABT and has been excluded from the relevant paragraph in the RM (Lines 316–328).

L273 - 'homologies of prokaryotes' is not appropriate wording. Could be 'homologous genes encoded by prokaryotes'. However, what/where is the evidence supporting this statement?

Response: Thanks for the suggestion. We have revised the related sentences as “However, phage P-gene became more important than prokaryotic P-gene in farmland, Gobi desert, and mine wasteland (Figure 6b, 6e, and 6f). In detail, the relative influences of phage P-gene in the three habitat types were 18% (farmland), 18% (Gobi desert), and 19% (mine wasteland), correspondingly.” (RM: Lines 325–328). This statement was supported by the results from the aggregated boosted tree (ABT) analysis (as shown in Figure 6b, 6e, and 6f in the RM).

L285 - Host analysis results need to be better reported. Provide more details in terms of the number of linkages and what kind of hosts they are; e.g. are they involved in P-acquisition functions? This analysis should also be integrated with the phylogenetic analyses reported; e.g. L211.

Response: Thanks for the suggestion. In order to better report host analysis results and integrate them with those from the phylogenetic analyses, the related contents have been rewritten as follows (RM: Lines 331–346): “Applying a new integrated phage-host prediction method (iPHoP), 35% (26/75) of the P-acquisition vOTUs identified in our metagenomes could be associated with a total of 28 hosts, which spanned 10 bacterial and one archaeal phyla (Figure 7a and Supplementary Table 17). Nine and seven of the predicted hosts were assigned to *Actinobacteriota* and *Proteobacteria*, respectively. Approximately 70% (18/26) of the vOTUs involved in host-virus linkages encoded *phoD*, *phoR*, and *ugpQ*, and they had a broad host range (Figure 7a). Remarkably, vOTU44 that encoded *ugpQ* was associated with hosts from both the Bacteria (*Actinobacteriota*) and Archaea (*Thermoproteota*) domains (Supplementary Table 17). Regardless of their host and habitat type, most of the UgpQ proteins encoded by the viral *ugpQ* genes recovered from our soil metagenomes clustered together in the phylogenetic tree (Supplementary Figures 26a). A similar pattern was observed for the phage-associated PhoD proteins (Supplementary Figures 26b). In contrast, those proteins encoded by other P-acquisition AMG kinds (i.e., PPa, PstA, PstB, PstC, PstS, PhoB, PhoR, and PhoU) tended to cluster separately with corresponding reference proteins of different bacterial phyla (Supplementary Figures 20, 27 and 28).”

L445 - isn't it the same trimming that is already performed by default by VS2?

Response: To avoid confusion, the related sentence has been rewritten as “Contigs longer than 10 kb were screened by VirSorter2 (v2.2.3; default parameters) and were further refined by identifying and removing potential host contaminants using CheckV (v0.9.0; default parameters).” in the RM (Line 519–521).

L446 - also list the criteria.

Response: Done as suggested (RM: Lines 540–546).

L547 - CD-HIT at nucleotide level?

Response: Yes, you are right.

L473 - geNomad could also be used for straightforward taxonomic classification.

Response: Done as suggested. Using GeNomad, up to 96% of the P-acquisition vOTUs (72/75) were classified to the *Heunggongvirae* kingdom, among which 53% (40/72) were of the *Caudoviricetes* class. However, only one of the *Heunggongvirae* vOTUs could be further classified to a specific viral family (i.e., *Herpesviridae*). In this context, the family-level and finer classification of the *Heunggongvirae* vOTUs has been performed alternatively with PhaGCN. For more details on the viral taxonomy profiles, please refer to RM: Lines 130–139.

L588 - host predictions could be done using also other lines of evidence to improve the number of linkages identified; e.g. <https://doi.org/10.1016/j.chom.2020.08.003> or <https://doi.org/10.1371/journal.pbio.3002083>

Response: According to your suggestion, we have used iPHoP (v1.3.3) for host prediction in the RM (Lines 717–725). Applying the new method, we did find more virus-host linkages. Overall, 35% (26/75) of the P-acquisition vOTUs identified in our metagenomes could be associated with a total of 28 hosts, which spanned 10 bacterial and one archaeal phyla. For more details on the virus-host linkages, please refer to RM: Lines 331–339.

Figure 4 ‘GC’ is missing from b, c, and d

Response: Thanks for pointing out the mistakes. The original Figure 4a-4c has been

revised considerably and the GC information has been deleted (RM: Figure 2).

Responses to Reviewer #3's comments

Reviewer #3 (Remarks to the Author):

This manuscript investigated P acquisition genes in a variety of environments across China, compared those genes found to published datasets, and experimentally validated the enzymatic activities of two P acquisition-related proteins encoded by AMGs. This work shows that 65% of the P-acquisition genes recovered in their metagenomes were detectable in published soil metagenome datasets, underscoring the ubiquity of these AMGs globally. Overall, the manuscript is intended to address an important research question that could be of interest to the general readers of Nature Communications. We listed some major and minor comments as follows.

Response: We thank this reviewer for acknowledging the merits of our manuscript. We have tried our best to address each of your comments. Below, please find our point-by-point responses to your comments. We really hope you will find that our manuscript is improved considerably.

Major comments:

1. Data interpretation:

a. Line 136-137, I would be more careful about the interpretation of lytic/lysogenic annotations from VIBRANT. These classifications are based on the presence of integrases or whether the viral sequences were identified as integrated into a bacterial contig (<https://github.com/AnantharamanLab/VIBRANT/issues/16>). All the viral contigs that are not classified as lysogenic lifestyle will be assigned as lytic. That could be misleading especially for the contig fragments assembled from soil metagenomes.

Response: Agreed. To be conservative, the related contents have been deleted in the

revised manuscript (RM).

b. Percent confidence of the Phyre2 results is about model confidence. The percent of coverage and identity are the main parameters suggesting how similar the query structures are to the reference structures in the database. The way that the results were introduced in lines 201-202 could be a bit misleading. Another question...Could the active sites be located on the modeled structures?

Response: Thanks for the comment. In order to avoid misleading, we have revised the relevant sentence as “The structural model prediction of proteins encoded by 18 representative P-acquisition AMGs (one gene was selected for *ppa* Family I, *ppa* Family II, and each of the other 16 AMG kinds, respectively) at Phyre2 showed 100% confidence, 16%–61% identity, and 47%–98% coverage (Figure 3, Supplementary Figure 19, and Supplementary Table 12).” (RM: Lines 211 – 215).

As to the question whether the active sites are located on the modeled structures, our results in the RM showed that most of the modeled structures of the proteins encoded by the P-acquisition AMGs possessed the active sites. The related sentences in the RM read as follows: “The active sites were located on most of the modeled structures, except those of PstC and YjbB. Specifically, the modeled PstC structure had only 19% identity and 61% coverage with that of the template – an ABC transporter (LpqY-SugABC) that translocates trehalose (Protein Data Bank ID: 7CAF). The modeled structure of YjbB had only 20% identity and 47% coverage with that of the template – a human citrate transporter (NaCT, Protein Data Bank ID: 7JSK). The active sites of the two protein templates may be quite different from those of our proteins.” (RM: Lines 215 – 221).

c. It is surprising to see a rarefaction curve of the viral sequences detected from soil metagenomes (even with the sampling effort in this study) has reached a plateau. Are the mVCs de-replicated? A total of 35552 viral contigs from 333 soil metagenomes result in an average of 106 viral contigs per sample. This may suggest a low recovery

rate in viral sequences and partially explain the rarefaction curve. It could be good to acknowledge this limitation.

Response: Thanks for the comment. In the original manuscript, the mVCs were de-replicated. By the way, according to the Reviewer #1's suggestion, we have used vOTUs instead of mVCs in all analyses of the RM. Nonetheless, as per your suggestion, we have discussed this limitation in the RM as follows: "The seemingly saturated rarefaction curve of P-acquisition vOTUs with sample size of our soil metagenomes (Figure 1b) can be at least partly explained by a low recovery rate in viral sequences from the metagenomes. Thus, a deeper metagenomic sequencing effort and/or an application of metaviromics will probably capture more novel kinds of AMGs encoded by P-acquisition phages." (RM: Lines 399 - 404).

d. Line 395, potentially overstating the role of P acquisition genes in virus-host co-evolution.

Response: According to the comment, two revisions have been made in the RM. On the one hand, we have turned down our voice by rewriting the related sentence as "In this way, chronic P scarcity likely has an important impact on viral evolution as well as that of their hosts." (RM: Lines 465 and 466). On the other hand, we have obtained some evidence supporting the notion by calculating the ratios of non-synonymous to synonymous nucleotide differences (dN/dS) for P-acquisition AMGs (as per the Reviewer #1's suggestion). The related evidence has been discussed as follows: "The possibility seems to be supported by our finding that the calculated dN/dS values for P-acquisition AMGs were < 1 (Supplementary Table 13). Such values suggest that the related phages in terrestrial habitats are under selective pressures to keep their P-acquisition AMGs functional, as otherwise deleterious mutations will result in the loss of these AMGs (Kieft et al, 2021)." (RM: Lines 466 - 470).

e. Recommend highlighting the experimentally validated the enzymatic activities of two P acquisition-related proteins more to emphasize the importance of the finding. It

would be helpful for readers for you to separate that into its own section, dividing results validated computationally from results validated experimentally, as it gets lost in that section.

Response: Thanks for the suggestion. In the RM, we have assigned the results of the experimental validation into a new independent subsection of the Results section and named it as “Functional validation of phage *ppa* and *phoD*” (RM: 233 – 259).

2. Methodology:

a. it is unclear how the taxonomy of mVCs was assigned. Based on the method section, it was assigned by blasting mVC proteins against the NCBI viral proteins with known taxonomy. So the assignment is based on a single protein hit even given not all proteins are phylogenetic markers? The protein sharing network (fig1d) also showed that viral contigs with different ‘assigned’ taxa were grouped into the same cluster. it may indicate some disagreement of taxonomic assignment using the protein hit method VS the protein sharing method.

Response: Agreed. The related contents of the taxonomic assignment based on vContact2 and blast against NCBI viral protein database have been excluded from the RM. Alternatively, we have used geNomad (v1.7.0; according to the Reviewer #2’ suggestion) and PhaGCN (v2.0) for viral taxonomic classification, adopting the updated nomenclature released by the International Committee on Taxonomy of Viruses (ICTV). The new results on viral taxonomy profiles are presented in RM: Lines 130–139.

b. AMGs are first annotated by general databases (e.g., KEGG) as they share sequence similarities to the non-viral metabolic genes. Then mapping the short metaG or metaT reads (though high-quality)... how confident these AMG-mapped reads are not from the non-viral metabolic genes? Have the authors cross-checked the mapping results against the non-viral genes? If it is not carefully handled, it will affect some of the main discussions using quantitative results.

Response: Thanks for the comment. In the RM, we have tried to handle the related

datasets in a more careful way. On the one hand, the metagenomic datasets have been analyzed using the method proposed by Jian et al. (2021), to avoid the potential effects of host-derived reads on the calculation of relative abundances of the P-acquisition AMGs (Jian et al., 2021, *The ISME Journal*, 15, 3094–3110). That is, the high-quality metagenomic reads were mapped to the nucleotide sequences of the P-acquisition vOTUs, and the relative abundances of P-acquisition vOTUs were used to represent their AMGs' relative abundances. For more details, please refer to RM: Lines 663–679.

On the other hand, the metatranscriptomic datasets have been analyzed with the method that was employed by several important works in the AMG research field (e.g., Chen et al., 2020, *Nature Microbiology*, 5, 1504–1515; Kieft et al., 2021, *Nature Communications*, 12, 3503). In a word, the metatranscriptomic reads were mapped against the P-acquisition AMG sequences with no mismatch. Some details on the related analyses were listed as follows: First, as proposed previously (Kieft et al., 2021, *Nature Communications*, 12, 3503), the phage-host gene pairs for P-acquisition genes recovered from our metagenomes were identified by constructing phylogenetic trees containing individual kinds of P-acquisition AMGs and their bacterial counterpart gene encoding proteins recovered from the same metagenome, wherein the nearest relative of a given AMG was thought to be originated from the host of the phage carrying that AMG. Then, the filtered reads of each metatranscriptome under investigation (ranging from 10^7 to 10^8 reads per sample) were mapped to the P-acquisition AMGs and their corresponding host genes with no mismatch (Bowtie2, --very-sensitive), respectively. Within the 32 metatranscriptomes analyzed in this study, a total of 267 reads were mapped to the P-acquisition AMGs. Through cross-checking the mapping results against the AMGs and their host genes, we found that only one of the 267 reads were mapped to both the AMGs and their host genes. Therefore, we are confident that these AMG-mapped metatranscriptomic reads are very unlikely from the non-viral metabolic genes. In the RM, we have explained more clearly how we analyzed the metatranscriptomic datasets (RM: Lines 728–742).

Minor line edits

3. Line 106, P is not the only growth limiting factor for bacteria.

Response: Revised as “Since P is one of the growth limiting factors for bacteria...” (RM: Line 93).

4. Lines 42 and 273, homologs instead of homologies.

Response: In the RM, the related sentence has been rewritten and the term “homology” has been deleted (RM: Lines 34 - 37 and Lines 325 - 328).

5. Grammar corrections in regards to the use of commas (see the Oxford Comma) necessary throughout the Results and Discussion section (Lines 148, 151, 197, 214, 226, 240, 246, 264, 332, etc)

Response: Corrected whenever necessary throughout the RM (e.g., Lines 70, 72, 87, 143, 144, 146, 147, 354, 398, and 406).

6. Line 109, use of the word "comment" not understandable, perhaps you meant “similar”?

Response: Yes, you are right. We have revised the word “comment” as “similar” (RM: Line 96).

7. Line 112, it may be not accurate to say, ‘only one study’. There are other soil studies that have reported the detection of P-related AMGs including *phoH*. It is also risky to claim a study is the only one as it takes some time for a manuscript to publish and we are not sure if there will be new papers out. Please consider paraphrasing the sentence to be more accurate and avoid the uncertainty.

Response: Agreed. In the RM, we have rephrased the sentence as “However, such phages remain poorly described. In 2022, three viral *phoD* genes were detected by Zheng *et al.* (2022) in industrial soils and eight viral *phoH* genes were recovered by Han *et al.* (2022) from agricultural soils. More recently, Huang *et al.* (2024) identified

six P-acquisition AMGs spanning three distinct functional genes (i.e., *phoA*, *phoB*, and *phoH*) in paddy soils.” (RM: Lines 98 – 102).

8. Line 142, ‘ambiguous Caudovirales’ is not a common way to describe a taxon. Please define it in front.

Response: In the RM, the term “ambiguous Caudovirales” has been deleted. Meanwhile, we have used geNomad (v1.7.0; according to the Reviewer #2’ suggestion) and PhaGCN (v2.0) for viral taxonomic classification in the RM, adopting the updated nomenclature released by the International Committee on Taxonomy of Viruses (ICTV). The new results on viral taxonomy profiles are presented in RM: Lines 130–139.

9. Line 144-145, please note that Siphoviridae, Myoviridae and Podoviridae have been abolished by the recent release of ICTV.

Response: Thanks for the reminder. As mentioned above, we have adopted the updated nomenclature released by the ICTV in the RM.

10. Fig2 plots data of total mVC or P-acquisition phages? Please make the figure legend consistent with the text in lines 149-154.

Response: Thanks for the reminder. Figure 2 in the original manuscript plots P-acquisition phages. In the RM, the figure legend has been revised as “Ubiquity and uniqueness of soil P-acquisition vOTUs”. By the way, according to the Reviewer #1’s suggestion, the figure has been moved to the supplementary (i.e., Supplementary Figure 22 in the RM).

11. Please correct the spelling of ‘i.e.’ throughout the manuscript. add ‘, ’after ‘i.e.’.

Response: Corrected as suggested throughout the RM (e.g., Lines 80, 140, and 147).

12. Line 172, grammar (a dramatic increase)

Response: Corrected (RM: Line 150).

13. Line 177, it is unclear what is the 'highest'. Perhaps the most prevalent? The highest occurrence across soil samples?

Response: Sorry for the confusion. In the RM, it has been clarified that the 'highest' means 'the highest number of gene sequences' (RM: Lines 145 and 146).

14. Line 186 grammar (arrangements)

Response: Corrected (RM: Line 174).

15. Figure 4: The branches colored in black represent the phylogenetic placement of the reference bacterial sequences? Please make it clear in the legend key or text. Some of the reference clades are too big to clearly visualize the distribution of the AMGs. please consider either reducing the number of reference sequences or collapsing some of these clades.

Response: Thanks for the suggestions. Yes, you are right that the branches colored in black represent the phylogenetic placement of the reference bacterial sequences. This point has been explained in the figure legend. Meanwhile, the number of reference sequences has been reduced for better visualization. By the way, the phylogenetic trees in Figure 4 of the original manuscript have been moved to the supplementary of the RM. So, please refer to Supplementary Figures 20 and 26 - 28 in the RM for details on the related revisions.

16. Line 208, syntax and grammar (completely lacking)

Response: In the RM, the Results section has been almost completely rewritten and the related sentence was deleted.

17. Line 216-217, are the two PPase families corresponding to the certain clades of the tree in fig. 4e?

Response: Yes, the two PPase families are corresponding to two separate clades of the

tree (RM: Supplementary Figure 20).

18. Line 225-227, were the numbers of the P-acquisition phages counted after removing the AMG proteins lacking the known active sites?

Response: In the RM, the numbers of the P-acquisition phages have been counted after removing the AMG proteins lacking the known active sites.

19. Line 227 spelling (and)

Response: In the RM, the related sentence has been deleted.

20. Line 247-250, the same P-acquisition mVCs and the P-related AMGs were also detected in the global soil metagenomes. I am wondering if these mVCs in the global dataset also carried the same P-related AMGs as they were detected in the dataset curated in this study. It may be interesting to test if soil viruses harbor the same AMGs regardless of the different sites/terrestrial systems.

Response: To address your concern, we have manually scrutinized the mapped P-acquisition vOTUs to further determine whether the AMGs on these vOTUs are also mapped by the reads from the same metagenome. Remarkably, eight kinds of the AMGs (i.e., *phoR*, *phoU*, *pit*, *pstA*, *pstB*, *pstC*, *ugpE*, and *ugpO*) encoded by six of the mapped vOTUs were also detected in the global soil metagenomes (Supplementary Table 16 in the RM), indicating that some soil phages could harbor the same AMGs regardless of the different terrestrial ecosystems. The related contents have been added to the RM (RM: Lines 305 - 313 and 688 - 690).

21. Line 269-271, separating the % from the land type made this hard to read. Suggest reordering (i.e. mine wasteland (19.4%), farmland (19.2%), etc.).

Response: Done as suggested (RM: Lines 327 and 328).

22. Line 287, please make it clear what the ‘transcript ratio of phage:host pairs’ is.

Transcript coverage across the whole genomes/contigs or the P-related gene homologs?

Response: The ‘transcript ratio of phage:host pairs’ was calculated based on transcript coverage across the P-related gene homologs. To avoid confusion, the related expression has been revised as “transcript ratios of phage:host gene pairs associated with inorganic P solubilization” (RM: Lines 364 and 365).

23. Line 304-305, as mentioned, it is not accurate (e.g., DOI: <https://doi.org/10.1128/msystems.00076-18>, <https://doi.org/10.1038/s41396-022-01188-w>, <https://doi.org/10.1038/s41396-022-01188-w> and more examples in sediments...)

Response: Agreed and thanked you for recommending us the references. After carefully reading these references, we found that: (1) The first reference (i.e., <https://doi.org/10.1128/msystems.00076-18>) focused on soil viruses in permafrost and their potential roles in ecosystem carbon processing, while the potential roles of the viruses in soil phosphorous cycling was not mentioned in the main text. In Table S2 (entitled “Virally encoded auxiliary metabolic genes and other genes of interest”) of the reference, only one P-acquisition gene (i.e., *phoH*) was listed. However, this gene was not marked by the authors as an AMG (instead was associated with a question mark). In such a situation, we preferred not to cite the reference. Despite this, other references reporting phage-encoded *phoH* have been cited in the RM (as mentioned below). (2) The second reference (i.e., <https://doi.org/10.1038/s41396-022-01188-w>) is a replicate of the third reference, addressing the virus-encoded metabolisms (i.e., carbon, nitrogen, phosphorous, and sulfur) in organochlorine contaminated soils. Within that study, only one kind of P-acquisition AMG (i.e., *phoD*) was detected and listed in Table S8 of the reference. In the RM, we have cited this reference (RM: Lines 99, 100, and 394 – 396).

During the revision stage of our manuscript, Huang and his colleagues published a paper that reported three kinds of P-acquisition AMGs (i.e., *phoA*, *phoH*, and *phoB*) from paddy soils (Huang *et al.*, 2024, *Soil Biology and Biochemistry*, 189, 109279; available online 13 December 2023). This paper has been also cited in the RM (RM:

Lines 101, 102, and 394 - 396).

Additionally, two recent papers reporting four kinds of P-acquisition AMGs (i.e., *ppa*, *phoD*, *yjbB*, and *phoH*) from sediments have been cited in the RM as well (RM: Lines 86 - 90). We hope that these revisions have made our related statements more accurate.

24. Lines 327 & 343, ubiquity

Response: In the RM, the related sentence has been completely rewritten.

25. Line 322, as PPases are mainly involved in DNA, RNA, and protein syntheses. How confident are they classified as AMGs? If the confident assignment is based on the position of the *ppa* genes away from the other nucleotide metabolism genes, have the authors observed a clear differentiation of the AMG *ppa* and non-AMG *ppa*?

Response: In this study and the previous study reporting phage-encoded *ppa* genes from sediments (Luo *et al.*, 2022, *Microbiome*, 10, 190), the confident assignment was not based on the *ppa* genes away from the other nucleotide metabolism genes. According to the Reviewer #1's suggestion, we have used DRAM-v in DRAM (v1.2.0) to validate potential P-acquisition AMGs located in the vOTUs recovered from our soil metagenomes. As to the AMG *ppa* genes identified in our vOTUs, they were associated with auxiliary scores of 1-3 and flanked by viral hallmark or viral-like genes on both sides (e.g., *ppa54* shown in Figure 2c and *ppa1* shown in Figure S21d), supporting their affiliations to viruses. Additionally, *ppa* genes were also detected in public viral genomes including the genome of one isolated phage GCA_002606105.1 (RM: Figure 2c). Therefore, we are confident that the AMG *ppa* genes are very unlikely from non-viral genomes. In order to avoid misleading, the related sentence "... and it is also ubiquitous in nature as byproducts in many important microbial anabolic pathways such as DNA, RNA and oligosaccharide biosynthesis" has been deleted in the RM.

26. Discussion section, be consistent with italicizing genes or not (lines 329-337 in

particular)

Response: Gene names have been italicized throughout the RM (e.g., RM: Lines 397, 398, 405, and 406).

27. 355 grammar, components

Response: Corrected (RM: Line 437).

28. Line 370, they rather than it

Response: In the RM, the related sentence has been deleted.

29. Lines 383-385, tenses of verbs, grammar

Response: Corrected (RM: Lines 457 - 459).

30. Line 390, arising.. or acquired

Response: In the RM, the related sentence has been deleted.

31. Lines 392-394. Reduced intracellular content of P? unclear what you are referring to in this sentence.

Response: In the RM, “reduced intracellular content” has been revised as “a reduced intracellular content of P” (RM: Lines 463 and 464).

32. Line 397, exerts

Response: In the RM, the related sentence has been deleted.

33. Line 401, increasingly.

Response: Corrected (RM: Line 478).

34. Line 404, scales

Response: Corrected (RM: Line 480).

35. Line 440-441, grammar check.

Response: Done as suggested (RM: Lines 515–517).

36. Line 459, please write out the three criteria.

Response: Done as suggested (RM: Lines 540–546).

37. Line 463, checkV database? IMG/VR is one of the public databases mentioned in line 156. Please double-check.

Response: Revised as “IMG/VR and NCBI GenBank databases” (RM: Line 575).

38. Please provide version info of each tool used (e.g., MAFFT and IQ-TREE).

Response: Done as suggested (e.g., RM: Lines 590–592).

Responses to Reviewer #4’s comments

Reviewer #4 (Remarks to the Author):

Response: Thank you for reviewing our manuscript and providing us the constructive suggestions.

References

Chen, L.X., Méheust, R., Crits-Christoph, A., et al. Large freshwater phages with the

- potential to augment aerobic methane oxidation. *Nature Microbiology* 5, 1504–1515 (2020).
- He, X.J., Augusto, L., Goll, D.S., et al. Global patterns and drivers of soil total phosphorus concentration. *Earth System Science Data* 13, 5831–5846 (2021).
- Huang, X., Zhou, Z., Liu H., et al. Soil nutrient conditions alter viral lifestyle strategy and potential function in phosphorous and nitrogen metabolisms. *Soil Biology and Biochemistry* 189, 109279 (2024).
- Jian, H., Yi, Y., Wang, J., et al. Diversity and distribution of viruses inhabiting the deepest ocean on Earth. *The ISME Journal* 15, 3094 – 3110 (2021).
- Kieft, K., Zhou, Z.C., Anderson, R.E., et al. Ecology of inorganic sulfur auxiliary metabolism in widespread bacteriophages. *Nature Communications* 12, 3503 (2021).
- Luo, X.Q., Wang, P., Li, J.L., et al. Viral community-wide auxiliary metabolic genes differ by lifestyles, habitats, and hosts. *Microbiome* 10, 190 (2022).

Reviewer #1 (Remarks to the Author):

In general, the authors have addressed all the comments and input as listed below:

1. The Methods section has significantly improved from the last version.
2. AMG validations appear sufficient, and providing the annotation tables would be very helpful for the reader to understand the context of the viruses carrying P-acquisition genes.
3. The quality of the figures and the captions has also improved.

New comments:

Regarding the methods describing "accumulative curves" analysis to obtain the relative abundance table, it is essential to specify the R package used and its corresponding function. Although seemingly trivial, explicitly stating this information is crucial for reproducibility.

I want the author to be aware of the terms 'accumulative curve analysis' (Figure 1b, lines 126-129) and 'rarefaction analysis' (lines 399-404). Understanding the distinction between these terms is crucial for interpreting the results. Accumulative curves visualize the cumulative number of observed taxa as the sampling effort increases, while rarefaction analysis standardizes sampling effort to compare diversity levels among samples with different sequencing depths or sample sizes.

Lines 239-240, where it is stated, "Second, the function of a regulatory gene (e.g., *phoR*) is very difficult to verify in an in-vitro system," could you provide further elaboration on this point?

Reviewer #2 (Remarks to the Author):

The authors have addressed my comments.

L Braga

Reviewer #3 (Remarks to the Author):

The authors have adequately addressed all the comments and we do not have further questions.

Reviewer #4 (Remarks to the Author):

"I co-reviewed this manuscript with one of the reviewers who provided the listed reports. This is part of the Nature Communications initiative to facilitate training in peer review and to provide appropriate recognition for Early Career Researchers who co-review manuscripts."

Responses to the Reviewers' comments

Responses to Reviewer #1's comments

Reviewer #1 (Remarks to the Author):

In general, the authors have addressed all the comments and input as listed below:

1. The Methods section has significantly improved from the last version.
2. AMG validations appear sufficient, and providing the annotation tables would be very helpful for the reader to understand the context of the viruses carrying P-acquisition genes.
3. The quality of the figures and the captions has also improved.

Response: We thank this reviewer for acknowledging the efforts that we have made to address the comments from the first round of peer review.

New comments:

Regarding the methods describing "accumulative curves" analysis to obtain the relative abundance table, it is essential to specify the R package used and its corresponding function. Although seemingly trivial, explicitly stating this information is crucial for reproducibility.

Response: Agreed. In the latest version of our manuscript, we have clarified that: (1) Accumulative curves of all viral operational taxonomic units (vOTUs) as well as P-acquisition vOTUs were generated by a custom R script utilizing the R packages `foreach` and `doParallel`; and (2) The script is available on GitHub (<https://doi.org/10.5281/zenodo.10746127>). For more details, please refer to Lines 965-972.

I want the author to be aware of the terms 'accumulative curve analysis' (Figure 1b, lines 126-129) and 'rarefaction analysis' (lines 399-404). Understanding the distinction

between these terms is crucial for interpreting the results. Accumulative curves visualize the cumulative number of observed taxa as the sampling effort increases, while rarefaction analysis standardizes sampling effort to compare diversity levels among samples with different sequencing depths or sample sizes.

Response: Thank you for pointing out the distinction between the terms of 'accumulative curve analysis' and 'rarefaction analysis'. In the latest version of our manuscript, we have corrected the misapplied word 'rarefaction' as 'accumulative' (Line 403).

Lines 239-240, where it is stated, " Second, the function of a regulatory gene (e.g., phoR) is very difficult to verify in an in-vitro system," could you provide further elaboration on this point?

Response: Yes, in the latest version of our manuscript, we have provided further elaboration on this point, which reads as follows: "...given that it involves the purification of both PhoR and PhoB proteins, and the verification of the autophosphorylation of PhoR and the phosphorylation of PhoB simultaneously." (Lines 243-245).

Responses to Reviewer #2's comments

Reviewer #2 (Remarks to the Author):

The authors have addressed my comments.

L Braga

Response: Thank you for reviewing our revised manuscript.

Responses to Reviewer #3's comments

Reviewer #3 (Remarks to the Author):

The authors have adequately addressed all the comments and we do not have further questions.

Response: Thank you for reviewing our revised manuscript.

Responses to Reviewer #4's comments

Reviewer #4 (Remarks to the Author):

"I co-reviewed this manuscript with one of the reviewers who provided the listed reports. This is part of the Nature Communications initiative to facilitate training in peer review and to provide appropriate recognition for Early Career Researchers who co-review manuscripts."

Response: Thank you for reviewing our revised manuscript.